# Discovering Latent Graphs with GFlowNets for Diverse Conditional Image Generation

**Bailey Trang**[1], **Parham Saremi**[4,5], **Alan Q. Wang**[2],
**Fangrui Huang**[1], **Zahra TehraniNasab**[4,5], **Amar Kumar**[4,5],
**Tal Arbel**[4,5], **Li Fei-Fei**[1], **Ehsan Adeli**[1,2,3] *

[1]Dept. of Computer Science, Stanford University, Stanford, CA, USA
[2]Dept. of Psychiatry and Behavioral Sciences, Stanford University, Stanford, CA, USA
[3]Dept. of Biomedical Data Science, Stanford University, Stanford, CA, USA
[4]Center for Intelligent Machines, McGill University, Montreal, QC, Canada
[5]MILA - Quebec AI institute, Montreal, QC, Canada
`{trangn, alanqw, fangruih, feifeili, eadeli}@stanford.edu`
`{parham.saremi, zahra.tehraninasab, amar.kumar}@mail.mcgill.ca`
`tal.arbel@mcgill.ca`

## Abstract

Capturing diversity is crucial in conditional and prompt-based image generation, particularly when conditions contain uncertainty that can lead to multiple plausible outputs. To generate diverse images reflecting this diversity, traditional methods often modify random seeds, making it difficult to discern meaningful differences between samples, or diversify the input prompt, which is limited in verbally interpretable diversity. We propose Rainbow, a novel conditional image generation framework, applicable to any pretrained conditional generative model, that addresses inherent condition/prompt uncertainty and generates diverse plausible images. Rainbow is based on a simple yet effective idea: decomposing the input condition into diverse latent representations, each capturing an aspect of the uncertainty and generating a distinct image. First, we integrate a latent graph, parameterized by Generative Flow Networks (GFlowNets), into the prompt representation computation. Second, leveraging GFlowNets' advanced graph sampling capabilities to capture uncertainty and output diverse trajectories over the graph, we produce multiple trajectories that collectively represent the input condition, leading to diverse condition representations and corresponding output images. Evaluations on natural image and medical image datasets demonstrate Rainbow's improvement in both diversity and fidelity across image synthesis, image generation, and counterfactual generation tasks.

## 1 Introduction

Conditional image generation produces novel images that adhere to given input prompts or conditions[2] like text [32, 75]. In real-world scenarios, an input prompt has inherent ambiguity, which may correspond to multiple plausible output images [7, 40, 76, 85], especially when prompts are abstract, high-level information. For example, an input text prompt describing a "sunset scene" could map to many valid images, differing in factors such as season, light control, and overall ambiance. Similarly, in medical imaging, brain magnetic resonance images (MRIs) of two patients of the same age and gender may nevertheless display variability in the structure of brain regions and patterns of intensity

---

*Corresponding author.

[2]We refer to "prompts" and "conditions" interchangeably.

despite having identical conditions due to subject-specific and medical scanner-specific details. In both cases, failing to address inherent uncertainty and capturing diversity in generative models can lead to suboptimal decision-making, misinterpretations, and generation collapse, where limited and uniform outputs fail to represent the necessary variability [13, 18, 38, 39, 58, 73].

Previous attempts at generating diverse images in conditional image generation models, such as using GANs [19], diffusion models [10, 24], and latent diffusion models [62] (LDMs), can be categorized into two main approaches: (1) Traditional methods typically rely on randomness; for example, repeating the generation process with different random seeds or varying the random noise on the same seeds in diffusion models [25, 34, 49, 51] to create multiple outputs. While these methods can produce non-identical images, they often fail to capture true diversity of choices and may exhibit inherent biases; (2) Another line of work involves diversifying and adding details to the input prompt verbally using a pretrained Large Language Model (LLM) such as ChatGPT [22, 60, 82]. Although this approach can enhance the richness of the generated content, it is confined to text-based conditions and relies on external LLM models and their own biases. Consequently, these strategies may not adequately address the inherent uncertainty of conditional image generation tasks. In addition, a more versatile approach is needed to handle multiple condition types. For instance, generating medical images conditioned on age, sex, diseases, or other medical details can enrich datasets in fields where data collection is costly and time-consuming, such as in 3D brain MRI or chest X-ray datasets.

Addressing these limitations, we introduce Rainbow, a novel conditional image generation framework designed to produce diverse and plausible images. Rainbow can be integrated into any pretrained conditional image generative model. The primary idea is to create multiple images simultaneously that capture uncertainty by collectively reflecting the input condition. To achieve this, we aim to generate diverse condition representations that encapsulate various aspects of the uncertainty inherent in the input condition within the latent space. Each representation produces a distinct output image while the pretrained generative models remain frozen or minimally modified. As a result, Rainbow delivers a range of high-quality images that comprehensively interpret the input prompt.

To achieve diverse condition latent representations that collectively reflect the input condition, we first construct a graph structure, called the *latent graph*, within the latent representation computation. Next, we utilize Generative Flow Networks (GFlowNets) [4, 5] to sample diverse trajectories over the graph collectively representing the input condition. Specifically, GFlowNets are designed to capture uncertainty in tasks with multiple possible outputs (multiple modes) by sampling diverse high-quality intermediate representations (e.g., trajectories over a graph) that lead to varied outputs, each representing one possible optimal outcome (one mode of the solution space). GFlowNets have been applied to many contexts, including molecule generation [4], gene regulatory networks [3, 48], and dropout masks [42]. In Rainbow, trajectories generated by GFlowNets collectively capture diverse interpretations of the input condition, leading to diverse condition latent representations, which are subsequently used to produce diverse output images.

Our contributions include:

- First, we introduce Rainbow, a novel conditional image generation framework that captures uncertainty and produces diverse images.

- Second, by discovering the diversity in condition latent representations, Rainbow is applicable to any pretrained conditional generative model, regardless of the condition type, and addresses the limitation of relying on randomness during generation.

- Third, our experiments on text- and non-text-based conditions across natural images and medical images (brain MRIs and chest X-rays) demonstrate Rainbow's improved capacity to capture uncertainty, generate diverse and plausible images, and benefit downstream tasks.

## 2 Preliminary

### 2.1 Generative Flow Networks (GFlowNets)

GFlowNets is a probabilistic model that samples diverse high-quality objects *i.e.* diverse trajectories of node/edge through a graph, where the likelihood of generating a trajectory $x$ is proportional to an unnormalized probability or reward $R(x) = e^{-\mathbb{E}(x)}$, with $\mathbb{E}$ denoting the expectation of some quantity of interest associated with $x$ [4, 5, 44]. GFlowNets samples a sequence of actions that modify

a compositional trajectory (*i.e.*, adding one edge to a trajectory), starting from a universal initial state and continues through successive modifications dictated by a trainable policy until it reaches a terminal state or achieves a specific graph sparsity. This policy is trained so that the probability of terminating the trajectory $x$ at a particular final state is proportional to the reward $R(x)$.

Specifically, GFlowNets operate on a graph $G = (S, A)$, where $S$ is the set of states and $A$ is the set of actions (transitions). The objective is to model a nonnegative flow $F : A \rightarrow \mathbb{R}_{\geq 0}$, which defines the unnormalized likelihood of taking action to transform from state $s$ to $s'$ [44]. To ensure correct sampling, the flow $F$ needs to satisfy certain constraints, such as the flow matching constraints [5].

**Flow Matching Constraints** is the core principle of GFlowNets, which enforces that for any intermediate states, the incoming flow equals the outgoing flow. For any state $s$, the state flow $F(s)$ is defined as the total flow through state $s$, and the edge flow $F(s' \rightarrow s)$ is the flow along transitions $s' \rightarrow s$; subsequently, the flow matching constraints is formulated as $F(s) = \sum_{(s' \rightarrow s) \in A} F(s' \rightarrow s) = \sum_{(s \rightarrow s'') \in A} F(s \rightarrow s'')$. The goal is to train the GFlowNets model so that the state flow at any terminal state $s_T$ that obtains object $x$ is proportional to the reward $F(s_T) \propto R(x)$.

### 2.1.1 Detailed Balance Objective

Detailed Balance Objective [5] (DBO) is one of the training objectives for GFlowNets, along with other approaches such as *flow matching* [5] and *trajectory balance objectives* [44]. DBO enforces consistency between forward and backward transitions while aligning terminal states with a reward function $R(x)$. Let $s_i$ denote the state at step $i$, where $s_0$ is the initial state and $s_n$ is the terminal state that yields object $x$ after $n$ steps; DBO defines the *forward policy* $P_F(s_i|s_{i-1}; \theta)$ parameterized by $\theta$ is the probability of transitioning to state $s_i$ from $s_{i-1}$, while the *backward policy* $P_B(s_{i-1}|s_i; \theta)$ models the reverse transition; The *state flow* $F_\theta(s)$, a scalar function, estimates the unnormalized likelihood of passing through state $s$. Subsequently, the DB loss combines two critical terms:

$$\mathcal{L}_{\mathrm{DB}}(x, R(x)) = \sum_{i=1}^{n-1} \left( \log \frac{F_\theta(s_{i-1}) P_F(s_i|s_{i-1}; \theta)}{F_\theta(s_i) P_B(s_{i-1}|s_i; \theta)} \right)^2 + \left( \log \frac{F_\theta(s_{n-1}) P_F(s_n|s_{n-1}; \theta)}{R(x)} \right)^2 . \quad (1)$$

Specifically, the first term ensures conservation of flow between consecutive states $s_{i-1}$ and $s_i$. By minimizing the squared log-ratio of forward and backward transitions, the loss enforces that the forward probability to transform $s_{i-1}$ to $s_i$ equals the backward probability to revert $s_i$ back to $s_{i-1}$. In addition, by weighting with their respective flows, this term guarantees that the net flow through any state transition is balanced. The second term aligns the final state $s_n$'s flow with reward $R(x)$.

### 2.1.2 Capturing Diversity with GFlowNets

GFlowNets promote diversity through three interconnected mechanisms grounded in their flow-matching constraint foundation. First, the terminal state flow alignment $F(s_T) \propto R(x)$ enforces proportional sampling where candidates $x$ (associated with terminal states $s_T$) are generated with probability $p(x) = \frac{F(s_T)}{Z} \propto R(x)$, preserving all reward modes unlike reinforcement learning's $\arg\max R(x)$ objective [4]. Second, the flow conservation constraint ensures global balance: at every non-terminal state $s \in S$, incoming flows from predecessors equal outgoing flows to successors, preventing preferential routing to dominant modes while maintaining non-zero probability for all viable paths [5]. Finally, the stochastic forward policy $P_F(s''|s; \theta) = \frac{F(s \rightarrow s'')}{F(s)}$, derived from normalized edge flows $F(s \rightarrow s'')$, enables amortized trajectory generation. Unlike Markov Chain Monte Carlo's (MCMC) local random walks, this allows direct jumps between distant modes (e.g., structurally distinct molecular graphs with comparable $R(x)$) through single-pass sampling via $P_F$ [4], bypassing MCMC's iterative transitions [44]. Together, these mechanisms ensure diverse high-reward candidates are sampled proportionally to their rewards while preserving exploration capacity across disconnected regions of the solution space.

## 2.2 Latent Diffusion Models

Training an LDM consists of two stages. In the first stage, an autoencoder is trained which learns to map each image $\mathbf{X}$ to a lower-dimensional latent embedding $\mathbf{z}$. Let $\mathcal{E}_I$ and $\mathcal{D}_I$ denote the image encoder and decoder making up the autoencoder, respectively. In the second stage, a (conditional) diffusion model is trained on the optimized latent embeddings $\mathbf{z} = \mathcal{E}_I(\mathbf{X})$. The generative process of

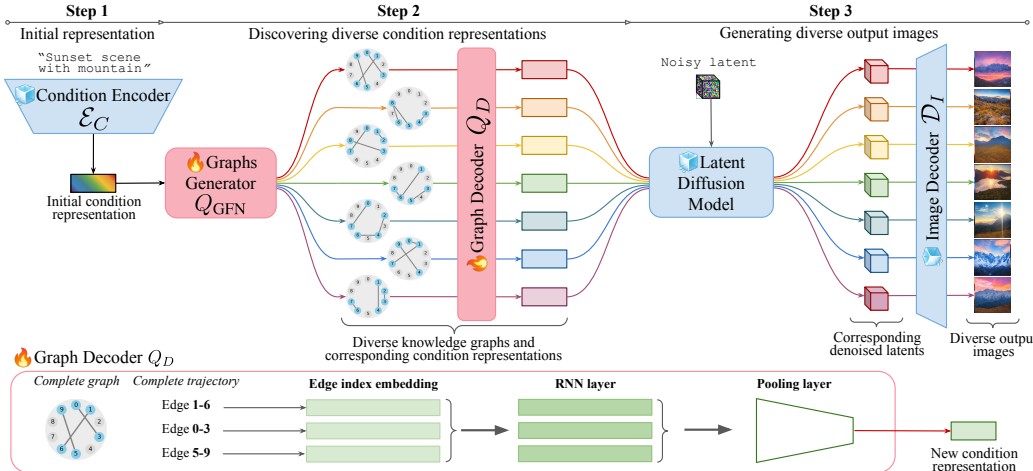

Figure 1: Rainbow operates by transforming an input condition into diverse images. Initially, it employs the pretrained condition encoder to derive an initial representation of the input condition (which contains uncertainty about locations or objects with the given prompt in this example). Then, a *graphs generator* produces multiple trajectories over a graph that reflect the input condition. These graphs are encoded into new latent condition representations. New condition representations and a latent noisy image are processed through the Latent Diffusion Model to acquire denoised image latents, which are subsequently decoded into diverse output images.

an LDM takes in a noisy latent $\mathbf{z}$ sampled from some prior distribution $p(\mathbf{z})$ and iteratively denoises it to produce a generated sample $\hat{\mathbf{z}}_0$ by $\hat{\mathbf{z}}_0 = \text{LDM}(\mathbf{z}, \mathbf{c})$, where $\mathbf{c}$ is the condition. Finally, the denoised latent is passed through the image decoder $\mathcal{D}_I$ by $\hat{\mathbf{X}} = \mathcal{D}_I(\hat{\mathbf{z}}_0)$ to obtain the synthesized image.

During training, noise is added to the latent representation to create a noisy latent image $\mathbf{z}_t$, where $t$ denotes the diffusion timestep. The model predicts the noise $\epsilon$ added to the latent image, minimizing the difference between the predicted noise $\hat{\epsilon}$ and the actual noise $\epsilon$ at every timestep $t$, as described in Equation 2, where $\epsilon_\omega$ is the neural backbone that performs time-conditioned denoising of the latent embedding. Typically, $\epsilon_\omega$ is implemented as a time-conditional UNet [63, 65].

$$\mathcal{L}_{\text{LDM}} = \mathbb{E}_{\mathcal{E}(\mathbf{X}), \epsilon \sim \mathcal{N}(0,1), t} ||\epsilon - \epsilon_\omega(\mathbf{z}_t, t, \mathbf{c})||_2^2. \tag{2}$$

## 3 Rainbow

Rainbow captures the inherent uncertainty in conditional image generation and produces diverse yet realistic images that reflect the input condition. The core objective is to decompose the input condition into diverse latent representations, distinct yet jointly interpret the same condition. Each new latent representation is then processed to generate an output image, enabling us to produce various images corresponding to the input condition. To achieve diversity in latent representations, Rainbow conducts a latent graph in latent representation computation and utilizes the GFlowNets [4, 5] to sample diverse high-quality trajectories over the graph, which are then decoded into condition latent representations to produce diverse output images.

### 3.1 Rainbow's Inference Pipeline

Let $M$ denote the number of images to be generated, and we assume that all models are fully trained. Figure 1 visualizes the inference process of Rainbow in three main steps.

**First**, the input condition (e.g. input prompt) $C$ is encoded into a condition initial latent representation, denoted as $\mathbf{c} \in \mathbb{R}^{S_c}$, (*e.g.* $\mathbb{R}^{77 \times 1024}$), via a learned condition encoder $\mathcal{E}_C$. **Next**, the graph generator $Q_{\text{GFN}}$ takes as input the initial condition embedding $\mathbf{c}$ and outputs a set of $M$ distinct trajectories over the graph. Each generated graph is then transformed into a new condition representation by the graph decoder model $Q_D$, as described in Equation 3. **Finally**, a latent diffusion model generates images from the set of condition embeddings $\hat{\mathbf{c}}^{1:M}$ and a set of noisy latents $\mathbf{z}^{1:M}$ sampled independently

from the prior to generate $M$ images $\hat{\mathbf{X}}^{1:M}$, described in Equation 4.

$$\hat{\mathbf{c}}^{1:M} = Q_D(Q_{\text{GFN}}(\mathbf{c})). \tag{3}$$

$$\left\{ \hat{\mathbf{X}}^i = \mathcal{D}_I(\text{LDM}(\mathbf{z}^i, \hat{\mathbf{c}}^i)), \ i = 1, ..., M \right\} \tag{4}$$

## 3.2  Rainbow's Training Details

This section describes the training strategy to obtain diverse condition latent representations $\hat{c}^{1:M} \in \mathbb{R}^{M \times S_c}$ from the initial $c \in \mathbb{R}^{S_c}$. A detailed algorithm is provided in Appendix C. We assume the existence of a pretrained LDM. During Rainbow training, we freeze condition encoders $\mathcal{E}_C$, image encoder $\mathcal{E}_I$, image decoder $\mathcal{D}_I$, and Unet model; the graph generator $Q_{\text{GFN}}$ and graph decoder $Q_D$ are trained from scratch. We present Rainbow training progress into three stages as detailed below.

**Stage 1: Discovering Diverse Graph Representations.**    We construct an undirected graph, $\mathcal{G}^*$, with $N$ nodes, which yields $N(N-1)/2$ non-self-loop edges. Inspired by previous works [47, 80] that learn the underlying connections of variables in the latent space for greater interpretable context exploration, edge embeddings in Rainbow are randomly initialized. During training, Rainbow assigns interpretable meaning to edges without being constrained by pre-defined edge semantics.

We design $Q_{\text{GFN}}$ as a GFlowNets model that iteratively predicts edges to be added to each of the set of $M$ trajectories over $\mathcal{G}^*$, while maintaining a fixed per-graph sparsity, $\rho$. At each step, the GFlowNets predicts $M$ edges, adding each edge to one of the $M$ trajectories. This process is repeated for $S$ steps. Specifically, the GFlowNets terminate the edge sampling process once $\rho$ is reached, and no explicit terminal states are defined (as in [48]). The total number of edges $S$ is calculated as $S = (1 - \rho) \cdot \frac{N(N-1)}{2}$. With this design, the number of states in the GFlowNets equals the number of edges plus 1, $S + 1$, which includes one initial state and $S$ states for adding $S$ edges.

In the initialization, we create a set of $M$ trajectories $\mathcal{T}_{s=0}^{1:M} = \{\tau_{s=0}^1, \tau_{s=0}^2, \ldots, \tau_{s=0}^M\}$, where $s$ represents the state index within the GFlowNets framework. We initialize each trajectory with a special starting element, the edge index 0. At state $s = S$, each trajectory in $\mathcal{T}_{s=S}^{1:M}$ is expected to be filled with $S$ *edge indices* in the range 1 to $\frac{N(N-1)}{2}$.

At states $s = i$ with $1 \leq i \leq S$, we input the previous trajectory $\mathcal{T}_{s=i-1}^{1:M}$ and the condition $\mathbf{c}$ to predict probability of edges to add to the current $M$ trajectories, one edge for each, as described in Equation 5. In Rainbow, the edge prediction strategy is derived from DBO, which is introduced in Section 2.1.1 with further details in Appendix C. After reaching the final state $s = S$, we obtain $M$ trajectories $\mathcal{T}_{s=S}^{1:M}$, each filled with $S$ edges.

$$\mathcal{T}_{s=i \in 1..S}^{1:M} = Q_{\text{GFN}}(\mathcal{T}_{s=i-1}^{1:M}, \mathbf{c}). \tag{5}$$

**Stage 2: Decoding Graphs into Condition Representations.**    A graph decoder model $Q_D$ is utilized to decode each trajectory into a condition representation of shape $S_c$, denoted as $\hat{\mathbf{c}}^{1:M} \in \mathbb{R}^{S_c}$. As visualized in Figure 1, $Q_D$ is designed with three key steps.

First, for each trajectory $\mathcal{T}_{s=S}^i$, $Q_D$ encodes the sequence of edge indices (of shape $\mathbb{R}^{S \times 1}$) into a sequence of edge embeddings of shape $\mathbb{R}^{S \times d_{\text{dim}}}$. Subsequently, these edge embeddings are passed through an RNN to ensure the order correlation of edges. Finally, the output of the RNN is processed by a projection pooling layer to map the sequence from $\mathbb{R}^{S \times d_{\text{dim}}}$ to the desired shape $\mathbb{R}^{S_c}$.

The final condition representations are computed as a convex combination between the diverse representations and the original condition representation $\mathbf{c}$ by a blending factor $\gamma$:

$$\hat{\mathbf{c}}^{1:M} = \gamma Q_D(\mathcal{T}_{s=S}^{1:M}) + (1 - \gamma)\mathbf{c}. \tag{6}$$

**Stage 3: Getting reward and computing losses.**    After obtaining $\hat{\mathbf{c}}^{1:M}$, we perform the diffusion process as introduced in Section 2.2 with the added noise $\epsilon_w$ and get $M$ predicted noise $\hat{\epsilon}^{1:M}$. Subsequently, to evaluate how *good* the sampled $M$ trajectories, corresponding to $M$ predicted noise, the reward function $R(\mathcal{T}_{s=S}^{1:M})$, defined as the exponential of the negative MSE, as in Equation 7.

Our training objective combines two loss terms. First, the *GFlowNets loss*, $\mathcal{L}_{\text{GFN}}$, follows the DBO introduced in Section 2.1.1) with reward function $R(\mathcal{T}_{s=S}^{1:M})$. Second, the *diffusion denoising loss*,

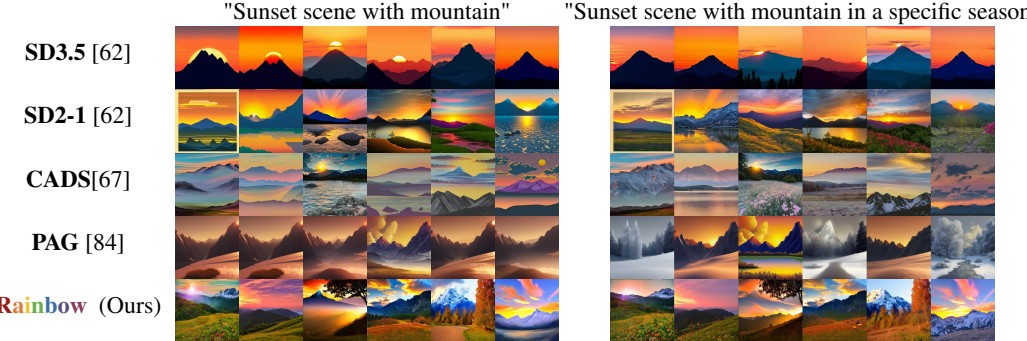

Figure 2: **Comparison of multiple images generated by baselines and Rainbow.** The baseline methods tend to produce images with repetitive layouts and primarily drawing art styles, failing to capture the uncertainty of "season". In contrast, Rainbow generates a variety of sunset scenes, showcasing diverse light levels, grass colors, and effectively capturing different seasons.

$\mathcal{L}_{\text{LDM}}$, (mentioned in Section 2.2) computes the mean squared error (MSE) between the added noise $\epsilon$ and the predicted noise $\hat{\epsilon}^{1:M}$. More specifically, $\mathcal{L}_{\text{GFN}}$ is to train the graphs generator $Q_{GFN}$, ensuring the diversity of sampled trajectories as well as the alignment to reward; meanwhile, $\mathcal{L}_{\text{LDM}}$ is to optimize the graph decoder model $Q_D$. The total loss $\mathcal{L}_{\text{total}}$ is a weighted combination:

$$R(\mathcal{T}_{s=S}^{1:M}) = e^{-\text{MSE}(\epsilon, \hat{\epsilon}^{1:M})}, \tag{7}$$

$$\mathcal{L}_{\text{GFN}} = \mathcal{L}_{\text{DB}}(\hat{\epsilon}^{1:M}, R(\hat{\epsilon}^{1:M})), \tag{8}$$

$$\mathcal{L}_{\text{total}} = \alpha \mathcal{L}_{\text{GFN}} + \beta \mathcal{L}_{\text{LDM}}. \tag{9}$$

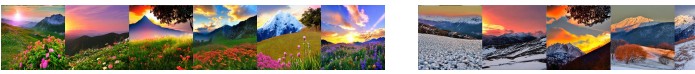

(a) Base prompt + **Spring** edges          (b) Base prompt + **Winter** edges

Figure 3: **Images generated by Rainbow seasonal edges** with the base prompt "Sunset scene with mountain". Most objects and layouts are consistent between the images, with noticeable season-specific details in the second image, such as spring flowers and winter snow.

## 4 Experiment

We conduct experiments to investigate the following hypotheses. $\mathcal{H}_1$: Utilizing diverse graphs facilitates generating diverse images; $\mathcal{H}_2$: Latent graphs can be extracted into meaningful and interpretable patterns; $\mathcal{H}_3$: Improved ability to capture diversity enhances the performance of downstream tasks. Reproducibility details are in Appendix D.

### 4.1 Experiment Setup

**Natural Images**    We use the Flickr30k dataset [83], which includes about 30k images paired with captions describing daily-life scenes, which contain uncertainty on object choices or styles. We build our Rainbow on top of the pretrained Stable Diffusion v2-1-base (SD2-1) with frozen pretrained VAE [35] image encoder/decoder, CLIP [56] text encoder, and Unet model. We evaluate the results against SD2-1, Stable Diffusion v3-medium (SD3.5), CADS [67] - a recent sampling strategy enhances diversity in the image-generation task, and pretrained checkpoint of PAG [84]- a recent work that improves diversity in the text-to-image task by prompt diversifying with GFlowNets. In our comparisons, both Rainbow and CADS utilize SD2-1's pretrained encoder-decoder and diffusion models. Rainbow's graph generator module includes $M = 40$, $N = 20$, and $S = 32$.

**3D Brain MRIs**    We curate a dataset of about 27k datapoints for training with no diagnosed disease from the following datasets: ADNI [53], ABCD Study [33, 78], HCP [77], PPMI [55], and AIBL [14]. This task contains uncertainty in anatomical details such as ventricle sizes. Our setting employs

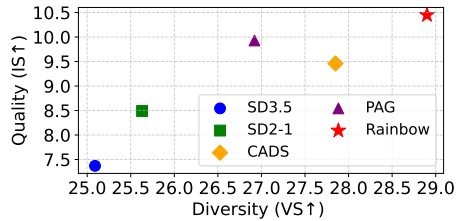
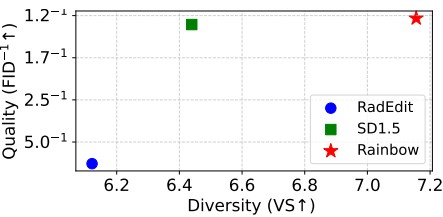

(a) Quantitative comparison on Natural Images

(b) Quantitative comparison on Chest X-rays

Figure 4: **Quantitative analysis on diversity and image quality of SD-based Rainbow**. Rainbow consistently outperforms SD baselines in diversity with higher Vendi Score (VS) across domains and image quality with higher Inception Score (IS) in natural images and $FID^{-1}$ in chest X-rays.

demographic input conditions on age and binary sex (0: male, 1: female) and fine-tunes the *Medical Open Network for Artificial Intelligence* (MONAI) [8]'s optimized 3D LDM along with Rainbow training. We benchmark Rainbow against LDM and a GAN-based [37] baselines. Rainbow's graph generator module includes $M = 8$, $N = 8$, and $S = 8$.

**Chest X-rays** We use the CheXpert dataset [30], which contains 170k training images. This dataset contains diversity in medical devices (such as chest tubes and wires), diseases (such as pneumonia and pleural-effusion) and anatomical details. We implement Rainbow on top of frozen parameters of a finetuned Stable Diffusion v1.5 (SD1.5) by previous work [36] for chest X-ray data. We generate 2D chest X-ray images based on text prompt conditions, *e.g., "Chest X-ray showing Support Devices"*. In addition to the finetuned SD1.5, we include RadEdit [52], a model trained from scratch on multiple chest radiology data such as CheXpert [29], MIMIC-CXR [31], and NIH-CXR [79] data for image editing tasks (more details at Appendix D.3), in the result comparison. Rainbow's graph generator module includes $M = 10$, $N = 20$, and $S = 33$.

## 4.2 Experiment Results

### 4.2.1 Diverse Images Generation Results

Investigating $\mathcal{H}_1$, we analyzed the generated images by baseline models and the proposed Rainbow in scenarios where the input condition contains uncertainty.

**Natural Images** Qualitatively, Figure 2 compares the generated images using two prompts, with 5 over 40 generations displayed. SD3.5 consistently generates repetitive layouts with backlit mountains, an orange sky, and ambiguous seasons. Additionally, SD2-1 offers a broader variety of objects and layouts but adheres to a drawing art style for the first prompt and predominantly generates spring scenes with green grass, lacking seasonal diversity for the second prompt. CADS marks diverse objects, yet produces unclear or winter-dominated images for the second prompt. PAG with one base image and diversified prompts does not introduce significant edits in this case and produces repeated layouts and many unclear season. Conversely, Rainbow produces images with diverse objects and light tones and effectively captures seasonal elements such as lush spring greenery and golden autumn foliage, even in the first prompt. For the second prompt, Rainbow demonstrates balanced seasons and clear seasonal features. We quantify the diversity and quality of generations in 60 prompts from the

| | FID score ↓ | | | Sex Classification Accuracy ↑ | | | Age MAE ↓ | | |
|---|---|---|---|---|---|---|---|---|---|
| **Real data** | 1e-5 | | | 96% | | | 3.55 | | |
| | Synthesis | Conv. age | Conv. sex | Synthesis | Conv. age | Conv. sex | Synthesis | Conv. age | Conv. sex |
| **Random** | - | - | - | | 48% | | | 32.17 | |
| **GAN** [19] | 2.3329 | - | - | 63% | - | - | 29.23 | - | - |
| **LDM** [8] | 0.3288 | 0.3570 | 0.3590 | 68% | 68% | 67% | 21.52 | 20.35 | 19.43 |
| **Rainbow** (Ours) | **0.3149** | **0.3375** | **0.3285** | **79%** | **80%** | **73%** | **13.73** | **18.59** | **16.40** |

Table 1: **Quantitative evaluation on 3D Brain MRIs.** "Conv." stands for "Converted". The lower FID score, the higher accuracy, and the lower mean absolute error (MAE in years) indicate better performance. Rainbow outperforms baselines across tasks and metrics.

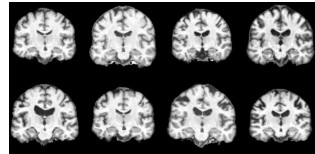 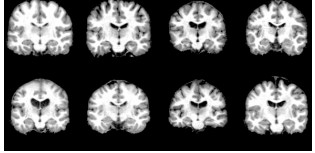 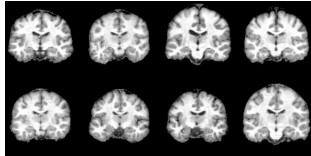

(a) Actual samples from dataset      (b) Generations by Rainbow      (c) Generations by baseline LDM

Figure 5: **Comparison of MRI image generations for *65-year-old male* individual.** Compared to actual samples from males aged 63-65, Rainbow captures greater diversity in details like ventricle sizes, while the baseline LDM generates images with less variation.

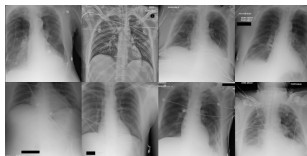 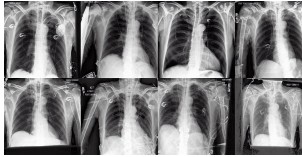 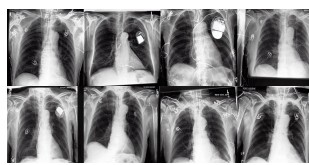

(a) Generations by RadEdit      (b) Generations by SD1.5      (c) Generations by Rainbow

Figure 6: Comparison of chest X-ray image generations for the prompt *Chest X-ray with support devices*. Rainbow is able to provide high-quality images while generating a more diverse set of medical devices compared to the baselines - (a) and (b).

COCO Validation set [41], with 40 images per prompt. We use the Inception Vendi score (VS) [16] to evaluate diversity and the Inception score (IS) [70] to assess image quality. As shown in Figure 4a, Rainbow outperforms the baseline in both diversity (higher VS) and image quality (higher IS). In addition, we observe that all methods produce images relevant to the prompts with a similar CLIP score [57], which is approximately 30.3. Numeric results are in Table 3, classifier-free-guidance scale affects are discussed in Figure 18, full 40 generations in Figures 12−16, all in Appendix E.

**Brain MRIs** Figure 5 showcases the generated brain MRI images conditioned on a 65-year-old male individual. Unlike younger age groups (*e.g.* 10-20 years old) with characteristic small ventricles, or the older age groups (*e.g.* 70+ years old) with large ventricles [1, 17, 71], the 60s age range includes a wide variety of ventricle patterns [61], as visualized in Figure 5a with actual samples. While all models generated high-quality images, Rainbow effectively captures the diverse ventricle patterns with varying ventricle sizes, whereas the baseline LDM tends to produce similar ventricle regions across different samples. We provide full axial, coronal, and sagittal views for this experiment in Figure 28 and visibility of structures and details in generated 3D MRIs in Figure 27 in Appendix E.

In addition, we quantify brain MRI generations obtained from multiple tasks: image synthesis (with 200 conditions balanced on age and sex) and counterfactuals on age (three age shifts at 10, 40, and 80 years) and sex (flipping binary sex). For each task, we report FID score [72] and performance on age and sex prediction by pretrained classifier models (details at Appendix D.2). As shown in Table 1, Rainbow outperforms LDM and GAN baselines across all metrics and tasks with lower FID, higher sex accuracy, and lower age MAE. To justify age and sex prediction models, we report "Real" results that were tested on real data and "Random" results that were evaluated on random outputs. Figure 29 in Appendix E provides counterfactual generations.

**Chest X-rays** Figure 4b quantifies generations by Rainbow and baselines using FID and VS. Rainbow achieves a higher VS, indicating greater diversity than the finetuned SD model, while also improving image quality with a lower FID score. Both Rainbow and SD outperform the RadEdit. Figure 6 provides a qualitative comparison, images are generated using the prompt *"Chest X-ray showing support devices"*, where Rainbow generates a more diverse set of medical devices, such as pacemakers, in all generations, while baselines do not show any devices in some images. All models achieve similar CLIP scores of 33.5. Additional results including generations, Figure 22 and numeric results, Table 4, are outlined in Appendix E.

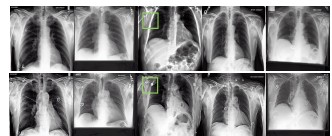 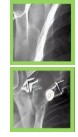 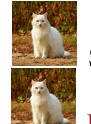 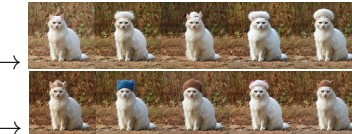

Figure 7: Images by Rainbow with the base prompt "Chest X-ray with no significant findings" (top) and being appended "supporting devices" edges (bottom) with devices added.

Figure 8: **Diversity in image editing**. Using the DAC [74] editing pipeline, given the cat (left) and "A cat wearing a wool cap", Rainbow captures diverse cap colors, while SD2-1 generates white caps.

#### 4.2.2 Latent Graph Interpretation

Investigating $\mathcal{H}_2$, we explore sets of edges that present for 4 seasons in Figure 2 Section 4.2.1. From images generated by Rainbow with the two prompts, we first cluster images based on observable seasonal features, *e.g.* snow for winter. We then extract the 10 most frequently added edges for each season when having "in a specific season" in the prompt. Subsequently, we append these 10 extra edges to the original trajectories of the first prompt and generate new images. Figure 3 presents the effect of manipulated graphs on the newly generated images. We can observe the addition patterns of colorful flowers in Figure 3a and snow in Figure 3b. Although edges in the latent graph are not predefined, Rainbow can implicitly learn to capture specific context and group edges into meaningful features. For images with four seasons' edges, see Figure 17 in Appendix E.

We apply the same approach to chest X-rays to explore the set of "support devices" edges by extracting the 10 most frequently added edges when changing the prompt from "Chest X-ray with no significant finding" to "Chest X-ray showing support devices" (visualized in Figure 25 in Appendix E). Figure 7 shows the transformation with added "support devices" edges into trajectories with the appearance of medical devices in the generated images.

#### 4.2.3 Performance on Downstream Tasks

Exploring $\mathcal{H}_3$, we conduct two downstream tasks: image editing and counterfactual generation.

We conduct an image-editing task with the natural-image domain that modifies the input image based on a prompt. We compare Rainbow to SD2-1 using the DAC image-editing pipeline [74]. Figure 8 shows the results of editing an image of a cat to add a wool cap. DAC with SD2-1 consistently produces images with a white cap, while Rainbow generates caps in multiple colors. This demonstrates Rainbow's ability to capture uncertainty (the cap color) and generate diverse samples. For the full 40 generations, see Figure 19 in Appendix E.

For 3D Brain MRIs illustrated in Figure 9, we perform the age prediction task using training data that includes synthesized data from Rainbow and the LDM baseline. We include 1600 generations arranged by age from 0 to 100 for both sexes, with each condition generating 8 samples. Real data is incorporated in specific proportions. Figure 9 visualizes that models trained with synthesized data generated by Rainbow achieved better performance. Both models performed best and better than models trained with fully real data at 50%. Specifically, at 50%, both models outperformed those trained with only 1600 real data points (100% dashed line).

### 4.3 Ablation studies

We assess the effect of varying the number of trajectories $M$. As shown in Figure 10 (left) for 3D brain MRI, a decrease in $M$ leads to a drop in performance, with a significant performance decrease when $M = 1$. However, even with $M = 1$, Rainbow still performs better than the baseline LDM. A similar pattern is observed in Figure 10 (right) for chest X-ray data; the lowest performance corresponds to the lowest number of graphs, given the same sparsity, with $M = 10$ yielding the best performance on diversity assessment. Models for synthesizing chest X-rays are partially trained for 16,500 steps (out of 24,000 steps for fully trained models) with a fixed $N = 20$. Additional ablation results varying the number of nodes are provided in Appendix E.2.

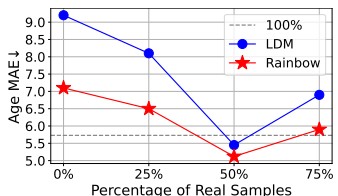

Figure 9: **Age Prediction Task trained on synthesized data.** Rainbow achieves the lowest MAE (Mean Absolute Error in years).

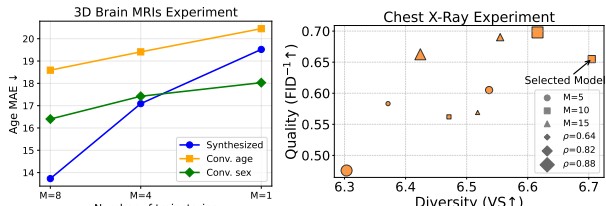

Figure 10: **Rainbow over number of trajectories ($M$)** showing that high values of $M$ yield better performance, which are MAE (Mean Absolute Error in years) on the 3D brain MRI and high diversity (VS score) in the Chest X-ray experiments. $\rho$ is the sparsity in Rainbow. "Conv." stands for "Converted".

## 5 Related Work

**Conditional Image Generation with Diffusion Models** have been driven by diffusion models, which iteratively denoise random inputs into coherent samples while outperforming GANs in stability and fidelity [26]. Key innovations include classifier-guided sampling [12], which steers generation using gradient signals from pretrained classifiers, and latent diffusion models (LDMs) [63], which operate in compressed latent spaces to enable high-resolution synthesis. Text-to-image models like DALL·E 2 [59] and Stable Diffusion [63] leverage large-scale multimodal pretraining to align textual prompts with visual concepts, while extensions like Palette [68] enable fine-grained control through spatial conditioning. These frameworks highlight the versatility of diffusion processes, with applications spanning artistic creation [69], medical imaging, and beyond.

**Diversity in Conditional Generation** Balancing diversity and fidelity remains a core challenge in conditional generation. GAN-based approaches address mode collapse via mode-seeking regularization [46] or self-conditioned clustering [43], while diffusion models inherently trade off diversity and quality through their noise schedules [2]. Methods like ControlNet [86] enhance controllability by injecting spatial constraints (e.g., edges, depth maps) into diffusion processes, whereas mutual information regularization [87] improves statistical dependency between latent codes and outputs. Adversarial training with semantic-guided negative sampling [9] further refines diversity in GANs, while category-consistent objectives [27] optimize photorealism and variation simultaneously. These advances collectively enable richer, more varied outputs without sacrificing semantic alignment.

**Generative Flow Networks** (GFlowNets) [4] offer a paradigm for sampling compositional objects (e.g., molecules, graphs) with probabilities proportional to a reward function, prioritizing diversity over the single-mode convergence of RL. Recent work extends this to sequential domains via recurrent architectures [50], demonstrating their capacity to model temporal dependencies in tasks like program synthesis. Our work adapts GFlowNets to *knowledge graph generation*, using the Trajectory Balance objective [45] to align forward edge-addition policies with backward inference while preserving order-dependent semantics through RNNs. To our knowledge, this is the first integration of GFlowNets with latent diffusion models.

## 6 Conclusion

We introduced Rainbow, a novel conditional image generation framework that captures uncertainty and produces diverse, plausible images. Rainbow constructs a latent graph in latent representation computation and leverages Generative Flow Networks to sample diverse trajectories over the graph, thereby enhancing the diversity of the condition latent representations and outputting diverse images that collectively interpret the input condition. Our experiments across natural and medical images not only demonstrate that Rainbow outperforms existing baselines in generating diverse, plausible images, but also highlight Rainbow's flexibility in adapting to any condition type. We discuss limitations and directions for future research in Appendix B.

## Acknowledgement

This work was supported in part by National Institutes of Health grant AG089169. This study was also supported by the Stanford School of Medicine Department of Psychiatry and Behavioral Sciences Jaswa Innovator Award, and the Stanford HAI Hoffman-Yee Award. Part of the data used in the preparation of this article was obtained from the Australian Imaging, Biomarker & Lifestyle Flagship Study of Ageing (AIBL) database (aibl.csiro.aug), with preprocessing and harmonization performed by Mind Data Hub at Stanford University. We thank Mohammad Abbasi from Stanford Translational AI (STAI) in Medicine and Mental Health Lab, Stanford University, for his contribution to preprocessing all the 3D brain MRI datasets used in this work. Additionally, we would like to thank Natural Sciences and Engineering Research Council of Canada, the Canadian Institute for Advanced Research (CIFAR) Artificial Intelligence Chairs program for grants; Mila - Quebec AI Institute, Google Research, Calcul Quebec, and the Digital Research Alliance of Canada for providing grants and computing resources.

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

# A  Broader Impacts

In this work, we presented Rainbow, a method that generates diverse images. These types of methods can play a crucial role in advancing AI models by enhancing robustness, reducing biases, and improving performance in various downstream tasks. By ensuring greater diversity in generated data, these models help mitigate biases that arise from underrepresented groups, building more equitable and generalizable AI systems. Additionally, generative models can be leveraged for counterfactual image generation, enabling the exploration of alternative scenarios for scientific and medical applications. However, while diversity is valuable, the realism of generated natural images raises concerns about misinformation and the misuse of synthetic content. In medical applications, counterfactual images must be rigorously validated by domain experts before being used to train autonomous systems, as incorrect or misleading data could lead to severe clinical consequences. Expert validation ensures that these images maintain diagnostic fidelity, preserving patient safety and the integrity of AI-driven medical decision-making.

# B  Limitations and Future Work

Although we introduced significant advancements in Rainbow and demonstrated experimental improvements, some limitations suggest potential areas for future work. One limitation is the higher computational resources required for training Rainbow, as we update $M$ trajectories in parallel. Future research could focus on optimizing the training process to alleviate these computational demands. Although working on latent graph reveals a high level of flexibility for Rainbow to be applicable to any kind of condition, another area for improvement is the interpretation of latent graphs in Rainbow. This would aid in enhancing the latent graphs' interpretability more automatically.

In terms of directions for future exploration, one direction is to extend Rainbow to other domains that require diversity and the ability to manage uncertainty, such as text generation, recommendation systems, and decision-making tasks. Another promising direction is to scale Rainbow to even larger latent graphs, which could capture uncertainties across multiple tasks. This expansion could lead to the creation of a foundational world model capable of addressing uncertainties in a variety of applications. One crucial analysis to add is anatomical plausibility tests and checks before making use of these images for any clinical application.

# C  Further Computation Details

## C.1  Conditional Image Generation with Latent Diffusion

**During Training**    The goal is to generate an output image $\mathbf{X}$ given an input image $\mathbf{I} \in \mathbb{R}^{\mathcal{S}_I}$ with shape $\mathcal{S}_I$ and a condition $\mathbf{C}$. Initially, the input image $\mathbf{I}$ is encoded into a latent representation $\mathbf{z}_0^I = \mathcal{E}_I(\mathbf{I})$ using an encoder $\mathcal{E}_I$. The latent code $\mathbf{z}_0^I \in \mathbb{R}^{\mathcal{L}_I}$ represents the underlying structure of the image in a lower-dimensional latent space, where $\mathcal{L}_I$ denotes the dimensions of this latent space.

The latent code $\mathbf{z}_0^I$ is then subjected to a forward diffusion process, which iteratively adds noise over $T$ steps:

$$q(\mathbf{z}_t^I \mid \mathbf{z}_{t-1}^I) = \mathcal{N}(\mathbf{z}_t^I \mid \sqrt{\alpha_t}\mathbf{z}_{t-1}^I, (1-\alpha_t)\mathbf{I}),$$

where $\alpha_t$ is a variance scheduling parameter that controls the amount of noise added at each step.

Concurrently, the condition $\mathbf{C}$ is encoded into its own latent representation $\mathbf{c} = \mathcal{E}_C(\mathbf{C})$ using an encoder $\mathcal{E}_C$. This latent representation $\mathbf{c} \in \mathbb{R}^{\mathcal{L}_C}$ encapsulates the conditioning information needed for the generation process, where $\mathcal{L}_C$ denotes the dimensions of the conditional latent space.

During the reverse diffusion process, the objective is to reconstruct the original latent representation $\mathbf{z}_0^I$ from the noisy latent code $\mathbf{z}_T^I$, guided by the condition latent representation $\mathbf{c}$. This reverse diffusion is generally modeled using a neural network $\epsilon_\theta$, which predicts the noise added at each timestep of the diffusion process in a commonly used formulation:

$$\mathbf{z}_{t-1}^I = \frac{1}{\sqrt{\alpha_t}}\left(\mathbf{z}_t^I - \frac{1-\alpha_t}{\sqrt{1-\alpha_t}}\epsilon_\theta(\mathbf{z}_t^I, t, \mathbf{c})\right) + \sigma_t\mathbf{n},$$

where $\mathbf{n} \sim \mathcal{N}(0, \mathbf{I})$ and $\sigma_t$ are scaling factors for the noise at step $t$. Note that there are variations in diffusion modeling where different parameter schedules or noise prediction methods might be employed.

Finally, the reconstructed latent code $\mathbf{z}_0^I$ is decoded back into the image space to produce the final output image $\hat{\mathbf{X}} = \mathcal{D}(\mathbf{z}_0^I)$, where $\mathcal{D}$ is the decoder function that maps the latent representation back to the high-dimensional image space, yielding the generated image $\hat{\mathbf{X}}$.

**Image Generation from Input Condition** Along with the encoded latent representation $\mathbf{c}$ from input condition $C$, a latent image $\mathbf{z}_T$ is sampled from a Gaussian prior distribution, typically $\mathbf{z}_T \sim \mathcal{N}(\mathbf{0}, \mathbf{I})$. The reverse diffusion process, conditioned on $\mathbf{c}$, iteratively refines $\mathbf{z}_T$ until obtaining $\mathbf{z}_0$. This final latent code $\mathbf{z}_0$ is then decoded using $\mathcal{D}$ to produce the output image $\hat{\mathbf{X}}$.

**Counterfactual Generation from Input Image and Condition** The input image $\mathbf{I}$ is encoded into its latent representation $\mathbf{z}_0^I = \mathcal{E}_I(\mathbf{I})$. The counterfactual condition $\mathbf{C}'$ is encoded into $\mathbf{c}' = \mathcal{E}_C(\mathbf{C}')$. The latent code $\mathbf{z}_0^I$ is perturbed with noise and reverse diffusion, guided by $\mathbf{c}'$, is applied to generate a new latent code $\mathbf{z}_0$. Finally, this modified latent code $\mathbf{z}_0$ is decoded using $\mathcal{D}$ to obtain the counterfactual image $\hat{\mathbf{X}}$.

## C.2 GFlowNets Training

The Algorithm 1 describes the training process of Rainbow to iteratively construct diverse trajectories over the latent graph using GFlowNets and the Detailed Balance (DB) objective. The process is divided into three phases: initialization, iterative edge sampling, and loss computation.

### C.2.1 Initialization Phase

The algorithm initializes with the input as the initial condition representation $c \in \mathbb{R}^{S_c}$. Configuration includes the number of parallel graphs $M$, the number of nodes $N$, and the sparsity $\rho$. The total number of edges $S$ is calculated as $S = (1 - \rho) \cdot \frac{N(N-1)}{2}$. A set of $M$ trajectories $\mathcal{T}_{s=0}^{1:M} \in \mathbb{R}^{M \times 1}$ is initialized with a special starting edge index $0$. Two masks—`forward_mask` $\in \mathbb{R}^{M \times n}$ and `backward_mask` $\in \mathbb{R}^{M \times n}$—are created to enforce valid transitions during sampling. These masks control which edges are available to be reached in the forward and backward paths. The log-likelihood difference tensor $ll\_diff \in \mathbb{R}^{(S+1) \times M}$ is initialized to track state transitions for the DB loss.

### C.2.2 Iterative Edge Sampling Phase

For each step $i \in \{1, \ldots, S\}$, the algorithm performs the following operations:

**Graph Encoding and Forward/Backward Probabilities**: The current trajectories $\mathcal{T}_{s=i-1}^{1:M}$ are encoded into latent representations $rep_g \in \mathbb{R}^{M \times h_g}$ using the graph decoder $\mathbf{Q_d^g}$. These are concatenated with the repeated condition $c$ to form a combined representation $rep \in \mathbb{R}^{M \times h}$. The forward predictor $\mathbf{MLP_{FW}}$ computes log-forward probabilities $log\_forward \in \mathbb{R}^{M \times n}$ and log-flow values $log\_flow \in \mathbb{R}^{M \times 1}$, masked to exclude already added edges. The backward predictor $\mathbf{MLP_{BW}}$ computes log-backward probabilities $log\_backward \in \mathbb{R}^{M \times n}$, masked to restrict invalid backward transitions.

**Edge Sampling and Mask Updates**: Edges are sampled from $log\_forward$ using a multinomial distribution, and the trajectories $\mathcal{T}_{s=i}^{1:M}$ are updated with the new edges. The log-likelihood differences $ll\_diff[i]$ accumulate the log-flow and log-forward probabilities of the sampled edges. For $i > 1$, $ll\_diff[i-1]$ is updated with backward probabilities to ensure transition consistency. The masks `forward_mask` and `backward_mask` are dynamically adjusted to reflect added edges.

### C.2.3 Finalization and Loss Computation Phase

After $S$ steps, the completed trajectories $\mathcal{T}_{s=S}^{1:M}$ are decoded into condition representations $\hat{c}^{1:M}$ using the graph decoder $Q_D$, blended with the original condition $c$ via the factor $\gamma$ as in Equation 6. The diffusion denoising process generates rewards $log\_reward \in \mathbb{R}^{M \times 1}$, which are incorporated into $ll\_diff[S]$. The DB loss $\mathcal{L}_{DB}$ is computed as the mean squared error of $ll\_diff$.

$$\mathcal{L}_{DB} = \text{mean}(ll\_diff^2).$$

**Algorithm 1** Rainbow Training Pipeline

---

1: **Input**: Intial condition representation $c \in \mathbb{R}^{S_c}$
2: $M$: Number of graphs that are computed parallelly
3: $N$: Number of nodes in graph
4: $\rho$: Sparsity
5: $n = N(N-1) * 0.5 * (1-\rho)$: Total number of edges in undirected graphs with $N$ nodes
6: $S$: Number of edges in the final graph
7:
8: **GFlowNets Architecture**
9: $h_g$, $h_c$: dimension of the encoded graph $g_i$ and the encoded condition $c$, respectively
10: $h = h_g + h_c$: GFlowNets hidden size
11: $\mathbf{Q_E^c} : \mathbb{R}^{S_c} \to \mathbb{R}^{h_c}$ $\qquad\qquad\qquad\qquad\qquad$ ▷ Condition encoder model
12: $\mathbf{Q_d^g} : \mathbb{R}^n \to \mathbb{R}^{h_g}$ $\qquad\qquad$ ▷ Instant graph decoder model (Used during edges sampling)
13: $\mathbf{MLP_{FW}} : \mathbb{R}^h \to \mathbb{R}^{n+1}$ ▷ Forward probability and flow predictor, $n$ dimensions for forward probability, and the last dimension for state flow
14: $\mathbf{MLP_{BW}} : \mathbb{R}^h \to \mathbb{R}^n$ $\qquad\qquad\qquad\qquad\qquad$ ▷ Backward probability predictor
15:
16: **GFlowNets computation flow for transforming state $i-1$ to state $i$**
17: **Inputs**: $\mathcal{T}_{s=i-1}^{1:M} \in \mathbb{R}^{M \times i}, \quad c' \in \mathbb{R}^{h_c}$
18: $rep_g = \mathbf{Q_E^g}(\mathcal{T}_{s=i-1}^{1:M}) \in \mathbb{R}^{M \times h_g}$ $\qquad\qquad$ ▷ Encoding graphs from state i-1
19: $rep = \text{concatenate}(rep_g, \ c'.repeat(M))$ $\qquad$ ▷ Concatenate encoded graphs and the repeated encoded condition
20: $pred = \mathbf{MLP_{FW}}(rep)$ $\qquad\qquad$ ▷ Preparing for forward probability and flow prediction
21: $log\_forward = \log\_\text{softmax}(pred[:,:-1] - forward\_mask * \inf)$ $\qquad$ ▷ Getting forward probability that with added edges excluded
22: $log\_flow = pred[:,-1:]$ $\qquad\qquad\qquad\qquad\qquad$ ▷ Getting log flow
23: $log\_backward = \log\_\text{softmax}(\mathbf{MLP_{BW}}(rep) - backward\_mask * \inf))$ $\qquad$ ▷ Getting backward probability among added edges
24: **Outputs**: $log\_forward \in \mathbb{R}^{M \times n}, log\_backward \in \mathbb{R}^{M \times n}, flow \in \mathbb{R}^{M \times 1}$
25:
26: **Step 0. Initialization**
27: $\mathcal{T}_{s=0}^{1:M} \leftarrow \mathbf{0}_{M,1}$ $\qquad\qquad$ ▷ Empty $M$ undirected graphs with a starting special edge index 0
28: $forward\_mask \leftarrow \mathbf{0}_{M \times n}, \quad backward\_mask \leftarrow \mathbf{1}_{M \times n}$ $\qquad$ ▷ Initialize masks
29: $ll\_diff \leftarrow \mathbf{0}_{(S+1) \times M}$ $\qquad$ ▷ First state for the initial state and $S$ states for adding $S$ edges, $ll\_diff[0]$ is not touched
30:
31: **Graphs generator training pipeline using GFlowNets and Detail-balance loss**
32: $c' = \mathbf{Q_E^c}(c)$
33: **for** $i$ in $1 \ldots S$ **do** $\qquad\qquad\qquad\qquad\qquad\qquad$ ▷ Loops of execution
34: $\qquad log\_forward, \ log\_backward, \ log\_flow = \mathbf{Q_{GFN}}(\mathcal{T}_{s=i-1}^{1:M}, c')$
35: $\qquad edges \leftarrow \text{multinomial}(log\_forward) \in \mathbb{R}^{M \times n}$ $\qquad$ ▷ Sampling edges as actions
36: $\qquad \mathcal{T}_{s=i}^{1:M} \leftarrow edges$ $\qquad\qquad\qquad$ ▷ Action as edges are added into trajectories
37:
38: $\qquad$ **# Updating flows**
39: $\qquad ll\_diff[i] += log\_flow + log\_forward.\text{gather}(actions)$
40: $\qquad$ **if** i > 1 **then**
41: $\qquad\qquad ll\_diff[i-1] -= log\_flow - log\_backward.\text{gather}(actions)$
42: $\qquad$ **end if**
43: $\qquad forward\_mask += actions$ $\qquad\qquad\qquad\qquad$ ▷ Updating the forward mask
44: $\qquad backward\_mask -= actions$ $\qquad\qquad\qquad\qquad$ ▷ Updating the backward mask
45: $\qquad$ **if** $i == S$ **then** $\qquad\qquad\qquad\qquad\qquad$ ▷ Completing the last turn
46: $\qquad\qquad \mathcal{T}_{s=S}^{1:M} = \mathcal{T}_{s=S}^{1:M}$
47: $\qquad\qquad \hat{c}^{1:M} = Q_D(\mathcal{T}_{s=S}^{1:M}) * \gamma + c * (1-\gamma)$ ▷ Decode *done* graphs into latent condition shape
48: $\qquad\qquad$ Performing Diffusion denoising conditioned on $\hat{c}^{1:M}$ and get $log\_reward \in \mathbb{R}^{M \times 1}$ as in Equation 8
49: $\qquad\qquad ll\_diff[S] -= log\_rewards$
50: $\qquad$ **end if**
51: **end for**
52: $\mathcal{L}_{DB} = ll\_diff^2.\text{mean}()$ $\qquad\qquad\qquad\qquad\qquad\qquad$ ▷ Optimization step

---

## D   Experiment Setup

Table 2 indicates hyperparameters used in this work for all experiments.

### D.1   Hyper-parameter

| Parameter Name | Natural Images | 3D Brain MRIs | Chest X-rays |
|:---:|:---:|:---:|:---:|
| Learning Rate | 1e-5 | 25e-7 | 1e-5 |
| Pretrained-LDM epochs | - | 80 | - |
| Rainbow epochs | 3 | 20 | 5 |
| Batch Size | 1 | 1 | 8 |
| $\alpha$ | 1 (Freeze Unet) | 0.2 | 1 (Freeze Unet) |
| $\beta$ | 1 | 0.8 | 1 |
| Training image shape | $3 \times 256 \times 256$ | $1 \times 160 \times 192 \times 176$ | $512 \times 512$ |
| Training condition type | Text prompt | Age and binary sexes | Text prompt |
| Training condition shape | Dynamic length | 2 dimensions | Dynamic length |
| Latent image shape | $4 \times 64 \times 64$ | $1 \times 32 \times 40 \times 48 \times 44$ | $4 \times 64 \times 64$ |
| Latent condition shape $S_c$ | $77 \times 1024$ | 256 | $77 \times 1024$ |
| Inference image shape | $3 \times 512 \times 512$ | $1 \times 160 \times 192 \times 176$ | $512 \times 512$ |
| Encoder - Decoder | VAE | VAE | VAE |
| Graph Size $N$ | 20 nodes | 8 nodes | 20 nodes |
| Number of Graphs $M$ | 40 | 8 | 10 |
| Sparsity $\rho$ | 0.83 | 0.70 | 0.82 |
| Num. Edges $S$ | 32 | 8 | 33 |
| Use RNN | Yes | Yes | Yes |
| Edge Embedding dim | 512 | 128 | 512 |
| Latent $dimh_g = h_c$ | 1024 | 1024 | 1024 |
| Blending factor $\gamma$ | 0.5 | 0.5 | 0.5 |

Table 2: Model Architecture and Parameter Indications

### D.2   Evaluation Metrics

**Natural Images**   To evaluate Rainbow on natural images, we use Inception Score (IS) and Inception Vendi Score (VS). For both metrics, we use the feature extraction model from pre-trained Pytorch Inception-v3.

**3D Brain MRIs**   We use Fréchet Inception Distance [23] (FID) to evaluate the feature quality of synthetic images. FID is a widely adopted metric for assessing the similarity between the feature distributions of real and generated images. It is based on the premise that high-quality synthetic images should exhibit feature distributions similar to those of real images when passed through a pre-trained neural network. FID is computed by calculating the Fréchet distance between two multivariate Gaussian distributions fitted to the feature vectors of real and generated images.

For feature extraction, we use a 3D ResNet50, which is particularly well-suited for capturing the complex 3D structures and patterns inherent in volumetric data. The model is trained on a diverse set of 23 medical imaging datasets, enabling it to generalize effectively across various medical image types, such as MRI and CT scans. The feature vectors are extracted from the final convolutional layer (conv seg), with a dimensionality of 2048. This layer captures high-level semantic features of the images, making it ideal for evaluating perceptual similarity between the real and synthetic images.

Lower FID values indicate that the distributions of real and generated images are more closely aligned, suggesting that the synthetic images are of higher quality.

To evaluate the faithfulness of capturing these conditions in our generated samples, we train a CNN-based age regressor and sex classifier on the real data (i.e., the same data that is used to train our proposed model and all baselines). The architectures for these CNNs can essentially be seen as the encoder half of a typical UNet [65], consisting of 4 downsampling levels and 2 convolutional blocks per level, with each block consisting of a convolution layer, batchnorm layer, and ReLU layer. The

age regressor is trained to minimize MSE loss, and the sex classifier is trained to minimize binary cross-entropy.

**Chest X-rays**    To evaluate Rainbow on chest X-rays, we use FID and Vendi Score similar to the previous modalities. For feature extraction, we use a pre-trained DenseNet-121 [28] model from the TorchXrayVision library [11], which is trained on multiple chest X-ray datasets such as CheXpert [29], NIH-CXR [79], PadChest [6], and MIMIC-CXR [31]. The feature vectors used for calculating the metrics are extracted from the last layer (before the classifier head) with a dimensionality of 1024.

## D.3    Baselines

**Natural Images**    We use 4 baselines: Stable Diffusion v2-1-base (SD2-1) [64][3], Stable Diffusion v3-medium [4], CADS [67], which is a sampling strategy that anneals the conditioning signal by adding scheduled, monotonically decreasing Gaussian noise to the conditioning vector during inference to balance diversity and condition alignment, and PAG [84], a novel approach that frames prompt adaptation as a probabilistic inference problem utilizing GFlowNet for the generation of diverse, high-quality prompts.

**Brain MRIs**    The GAN based model is from [37]. This 3D GAN model addresses both image blurriness and mode collapse problems by leveraging $\alpha$-GAN [66] that combines the advantages of Variational Auto-Encoder (VAE) and GAN with an additional code discriminator network. The model also uses the Wasserstein GAN with Gradient Penalty (WGAN-GP) loss [20] to lower the training instability.

The standard LDM is based on [54, 8, 63]. It can be viewed as identical to Rainbow, except that it does not leverage any latent graphs.

**Chest X-rays**    We consider two chest X-ray baseline models. The first model is RadEdit [52], a latent diffusion model developed by Microsoft Health. This model is trained on 487,680 frontal view chest X-rays of multiple datasets such as MIMIC-CXR [31], NIH-CXR [79], and CheXpert [29]. The second baseline is a fine-tuned Stable Diffusion v1.5 model [36] on the CheXpert [29] dataset.

## D.4    Training Time and Computation Sources

For general-domain experiments, training was conducted on a single NVIDIA H100-80GB GPU, completing in 12 hours.

The brain MRI experiment utilized 4 NVIDIA H100-80GB GPUs paired with 32 CPU cores over 3 days pretraining, followed by continued Rainbow training on the same hardware configuration for an additional 24 hours.

The chest X-ray experiments utilized 4 NVIDIA A100-80GB GPUS paired with 24 CPU cores. This model was finetuned on $512 \times 512$ chest X-rays and prompt pairs for 15 hours with an overall batch-size of 8.

# E    Addition Experiment Results

## E.1    Numeric Evaluation Results

Table 3 presents numeric results for Figure 4a.

Table 4 presents numeric results for Figure 4b.

## E.2    Chest X-Ray Ablation Studies

Figure 11 shows the results for the full ablation study across three parameters $N$, $M$, and $\rho$ for the chest X-ray model. The ablation setup is same as discussed in the main text, where each model is trained on a single GPU for 16500 steps. It can be observed that the models with more diversity (higher VS) have higher sparsity values, and the models with lower FID have lower sparsity. However,

---

[3]https://huggingface.co/stabilityai/stable-diffusion-2-1-base
[4]https://huggingface.co/stabilityai/stable-diffusion-3-medium

| Model name | IS ↑ | CLIP ↑ | Pixcel VS ↑ | Inception VS ↑ |
|---|---|---|---|---|
| SD3.5 | 7.37 | 30.29 | 3.75 | 25.09 |
| SD2-1 | 8.49 | 30.27 | 3.74 | 25.63 |
| CADS | 9.46 | 30.27 | 3.88 | 27.85 |
| PAG | 9.93 | 30.21 | 3.90 | 26.92 |
| DDIM | 8.44 | 30.28 | 3.91 | 26.62 |
| **Rainbow** | **10.45** | **30.32** | **3.94** | **28.90** |

Table 3: **Quantitative comparison of natural-image experiment**. An upward arrow "↑" indicates that a higher value corresponds to better performance. Rainbow consistently outperforms SD baselines in diversity (higher Vendi scores), image quality (higher IS score), and prompt context delivery (higher or comparable CLIP score)

| Model name | FID ↓ | CLIP ↑ | VS ↑ |
|---|---|---|---|
| **RadEdit** | 10.28 | **33.58** | 6.12 |
| **SD1.5** | 1.32 | 33.51 | 6.44 |
| **Rainbow** | **1.27** | 33.45 | **7.16** |

Table 4: **Quantitative comparison of Chest X-Ray experiment** . An upward arrow "↑" indicates that a higher value corresponds to better performance. Rainbow is increasing the VS and achieves higher diversity while improving the generation quality by lowering FID. All models have close CLIP similarity scores.

this does not imply that sparsity parameters are the single factor affecting quality and diversity. In general, we see more squares and triangles in the upper right section of the plot, showing that $M = 10$ and $M = 15$ are overall better than the $M = 5$ models. Furthermore, blue and orange are also more prevalent in the upper-right section of the plot indicating that models with $N = 25$ do not perform as good as models with fewer nodes.

### E.3 Full of 40 Generations for General-domain Experiments

Figure 19 provides the 40 generations that support Figure 8 in the main text.

Supporting Figure 2 in the main text, Figure 12 present full 40 generations with two prompts by SD3.5, Figure 13 present whole 40 generation with two prompts by SD2-1, Figure 14 present full 40

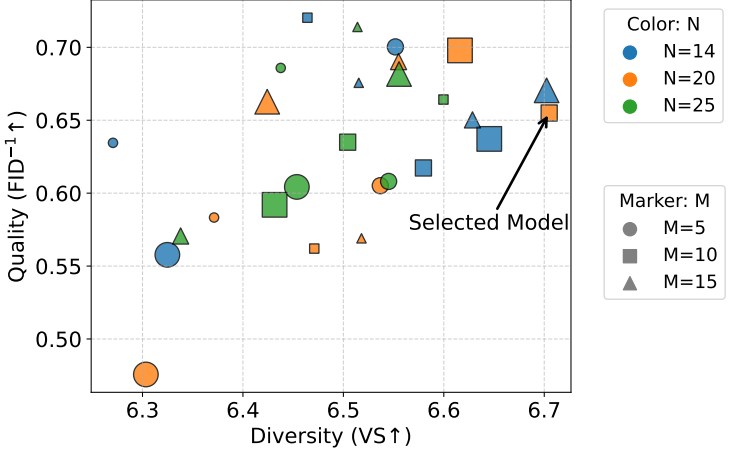

Figure 11: Ablation studies on the Chest X-Ray model by varying the number of nodes N, the number of graphs M, and the sparsity parameters $\rho$. Different colors show the N values (blue=15, orange=20, green=25). Different shapes show the M values (circle=5, square=10, triangle=15), and the size of the shapes shows the sparsity (biggest=0.88, middle=0.82, smallest=0.64). The final selected model is shown using the arrow labeled as 'Selected Model'.

generation with two prompts by CADS, Figure 15 present full 40 generation with two prompts by PAG, and Figure 16 present whole 40 generation with two prompts by Rainbow.

Supporting Figure 3 in the main text, Figure 17 presents full 40 generations with 4 sets of seasonal edges.

Figure 20 provides a qualitative and quantitative comparison of generated images between models with additional prompts.

Supporting Figure 6 in the main text, Figure 22 presents full 40 generations with 4 sets of seasonal edges.

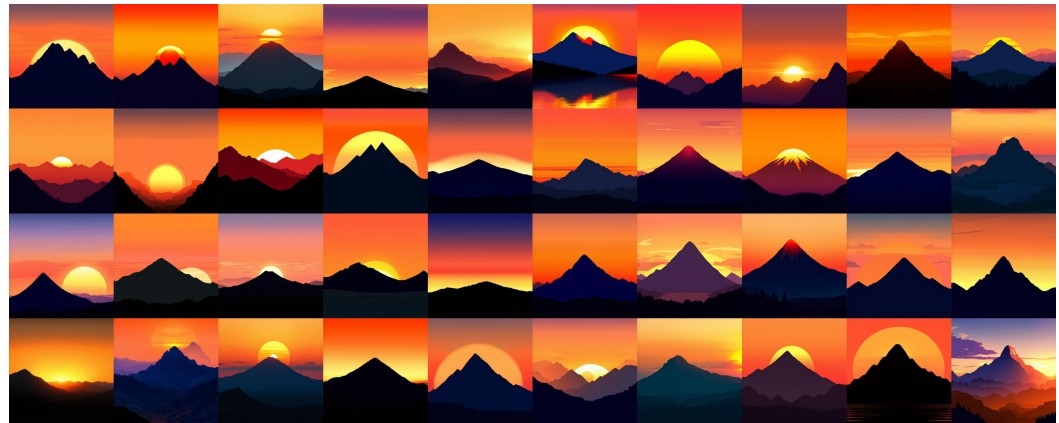

(a) "Sunset scene with mountain" by SD3.5

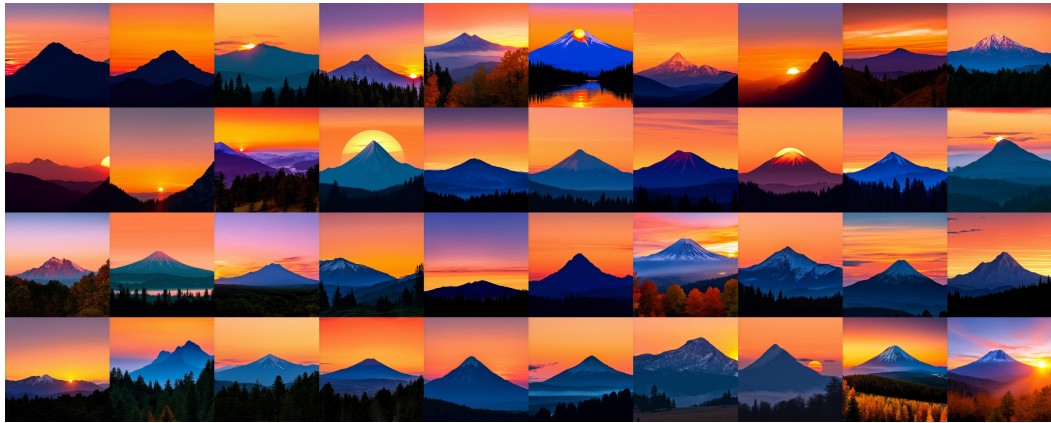

(b) "Sunset scene with mountain in a specific season" by SD3.5

Figure 12: SD3.5 generates repeated layouts and objects in both prompts, producing unclear seasons or dominated late-autumn/early-winter aesthetics in the second prompt.

### E.4 Addition prompts for Text-to-image task

Figure 20 provides a comparison of diverse natural-image generation between models in 4 additional prompts.

### E.5 3D Brain MRI Fidelity Analysis

As visualized in Figure 27, Rainbow maintains sharper anatomical details across all age groups while avoiding artifacts. **LDM** introduces subtle distortions, particularly in age-sensitive regions:

- **Young (14 vs.16):** Rainbow preserves fine textures like developing white matter tracts. LDM's output shows mildly blurred cortical layers.

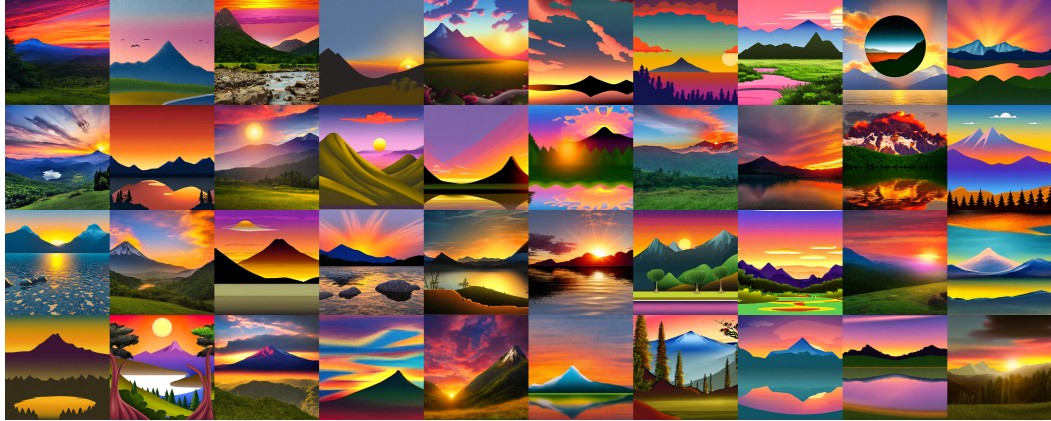

(a) "Sunset scene with mountain" by SD2-1

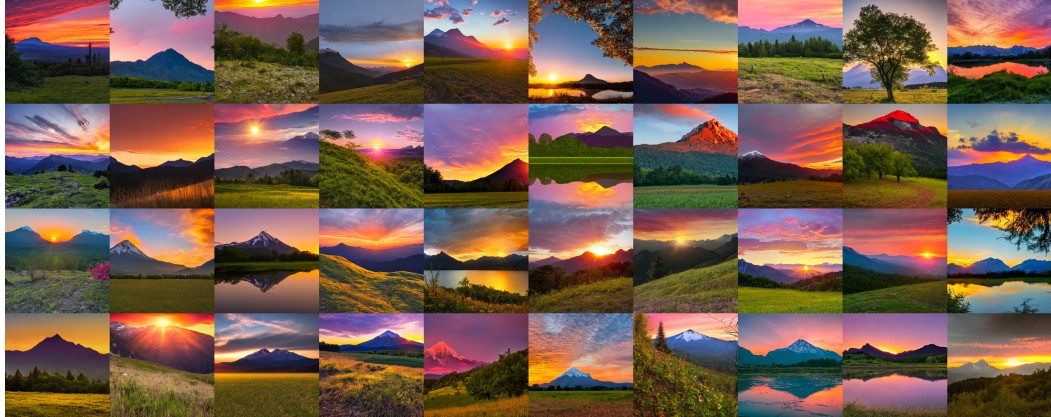

(b) "Sunset scene with mountain in a specific season" by SD2-1

Figure 13: SD2-1 generates diverse layouts and objects but heavily prioritizes a specific art style in the first prompt, while producing ambiguous or spring-dominated seasons in the second prompt.

- **Middle-aged (45 vs.44):** Rainbow retains small vessels and tissue gradients. LDM exhibits "smeared" edges around ventricles.
- **Elderly (75 vs.72):** Rainbow realistically renders age-related atrophy (e.g., widened sulci). LDM generates unnaturally smooth brain surfaces, masking thinning cortex.

### E.6 Qualitative Results for 3D MRI Counterfactual Generation Task

For brain MRI, we perform a counterfactual generation task. Given the factual image and condition, the task is to generate a counterfactual MRI based on a counterfactual condition on age or sex. Table 1 presents the numerical evaluation of the classification of sex and age based on the generated counterfactuals. Rainbow achieves higher performance, which underscores its effectiveness. Figure 29 compares the counterfactual generations. Both Rainbow and LDM generate correct patterns, such as smaller ventricles at a younger age (do(age=10)), evident from the red regions in the difference plot, and larger ventricles and cortex at an older age (do(age=80)), shown by the green regions. Additionally, sex conversion from male to female shows smaller ventricles and partially larger cortex regions. However, the baseline LDM exhibits some artifacts. These findings are consistent with previous studies on the effects of age and sex on MRI characteristics [15, 21, 81].

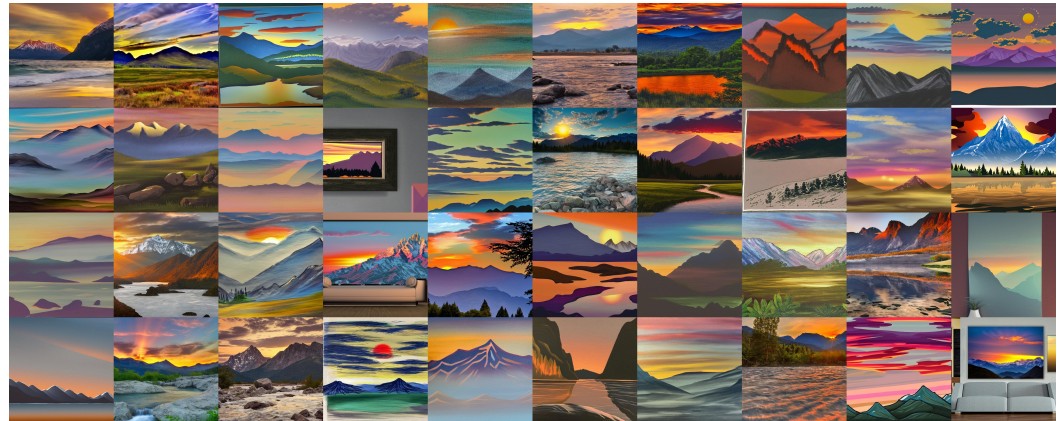

(a) "Sunset scene with mountain" by CADS

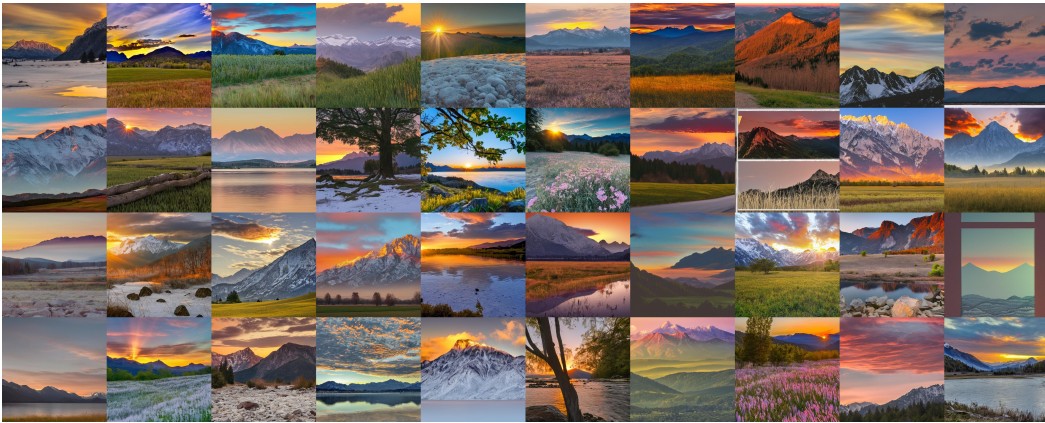

(b) "Sunset scene with mountain in a specific season" by CADS

Figure 14: CADS generates diverse layouts and objects, but prioritizes a specific art style in the first prompt. While CADS produces more defined seasonal characteristics in the second prompt, including generations with clear spring environments, some outputs retain ambiguity or disproportionately favor winter season.

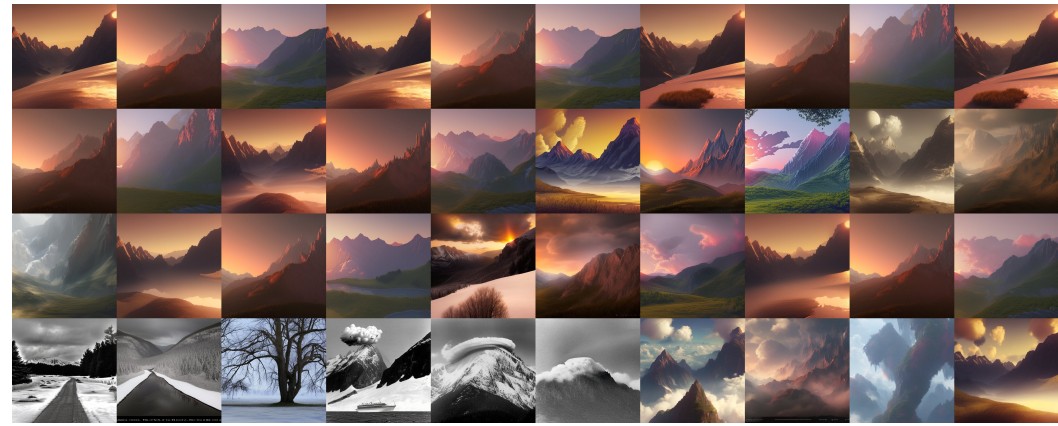

(a) "Sunset scene with mountain" by PAG

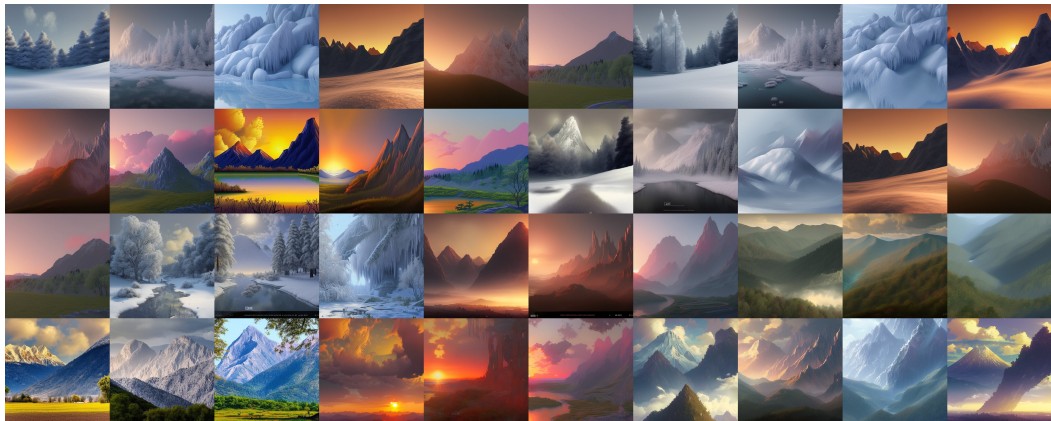

(b) "Sunset scene with mountain in a specific season" by PAG

Figure 15: Generations by PAG

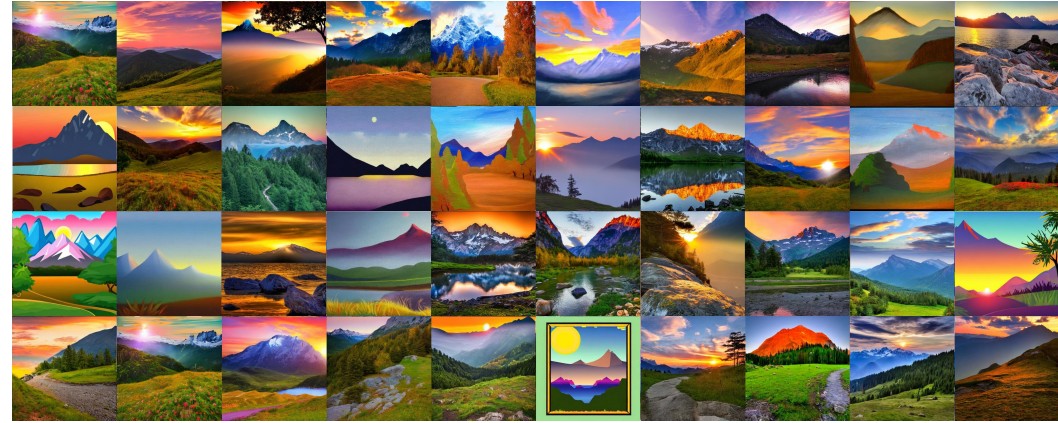

(a) "Sunset scene with mountain" by Rainbow

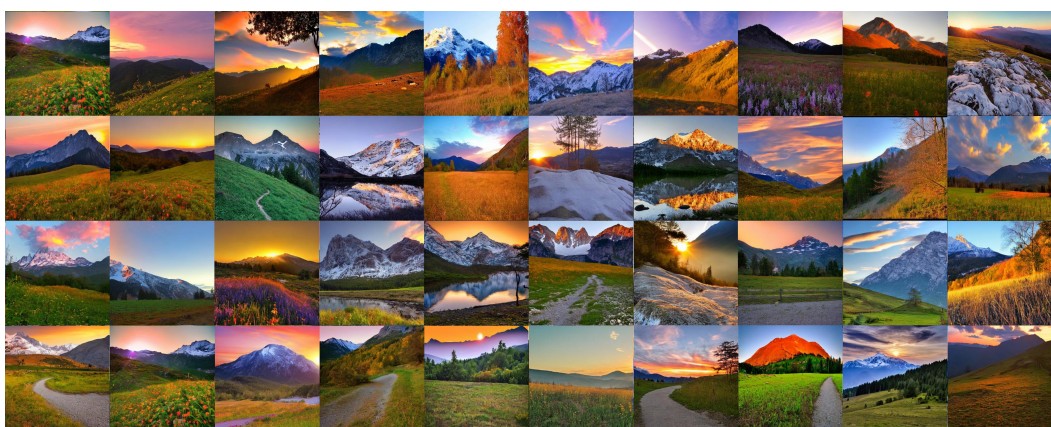

(b) "Sunset scene with mountain in a specific season" by Rainbow

Figure 16: Rainbow generates diverse layouts, objects, and seasonal environments with high compositional flexibility, achieving balanced variation across spring, summer, autumn, and winter visual details.

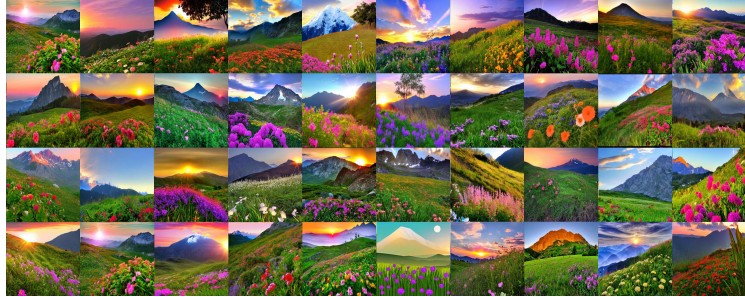

(a) "Sunset scene with mountain" + **Spring** edges

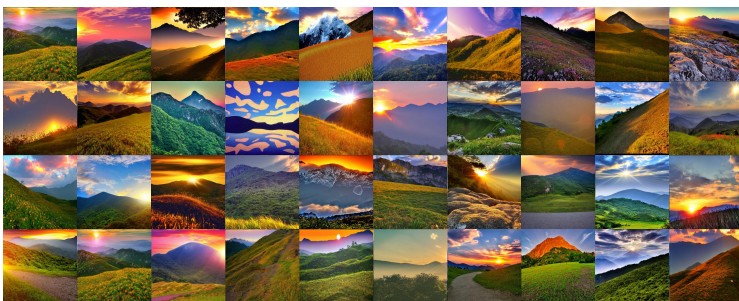

(b) "Sunset scene with mountain" + **Summer** edges

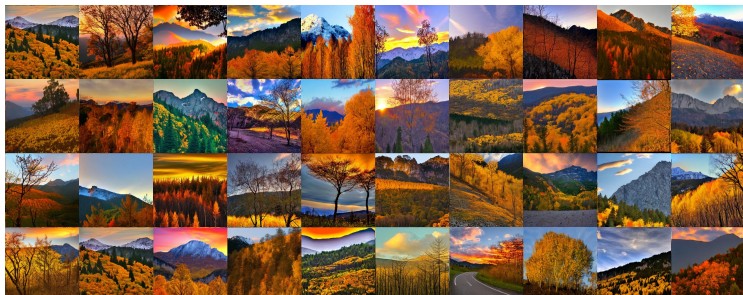

(c) "Sunset scene with mountain" + **Autumn** edges

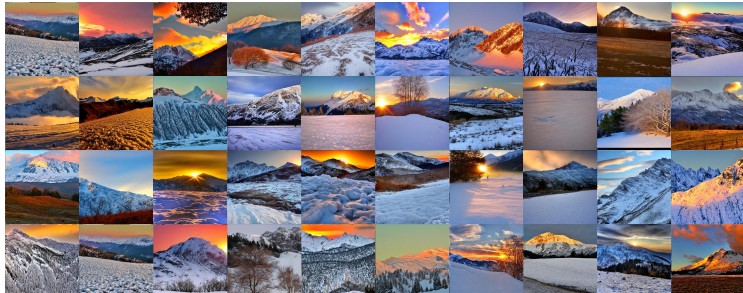

(d) "Sunset scene with mountain" + **Winter** edges

Figure 17: Comparison of images generated by Rainbow with manipulated trajectories with 10 extra edges specific to each season. We observe that objects and layouts are consistent between the images, with clear season-specific details such as colorful spring flowers and green grass, hot-toned sky for summer, autumn yellow leaves, and snow for winter.

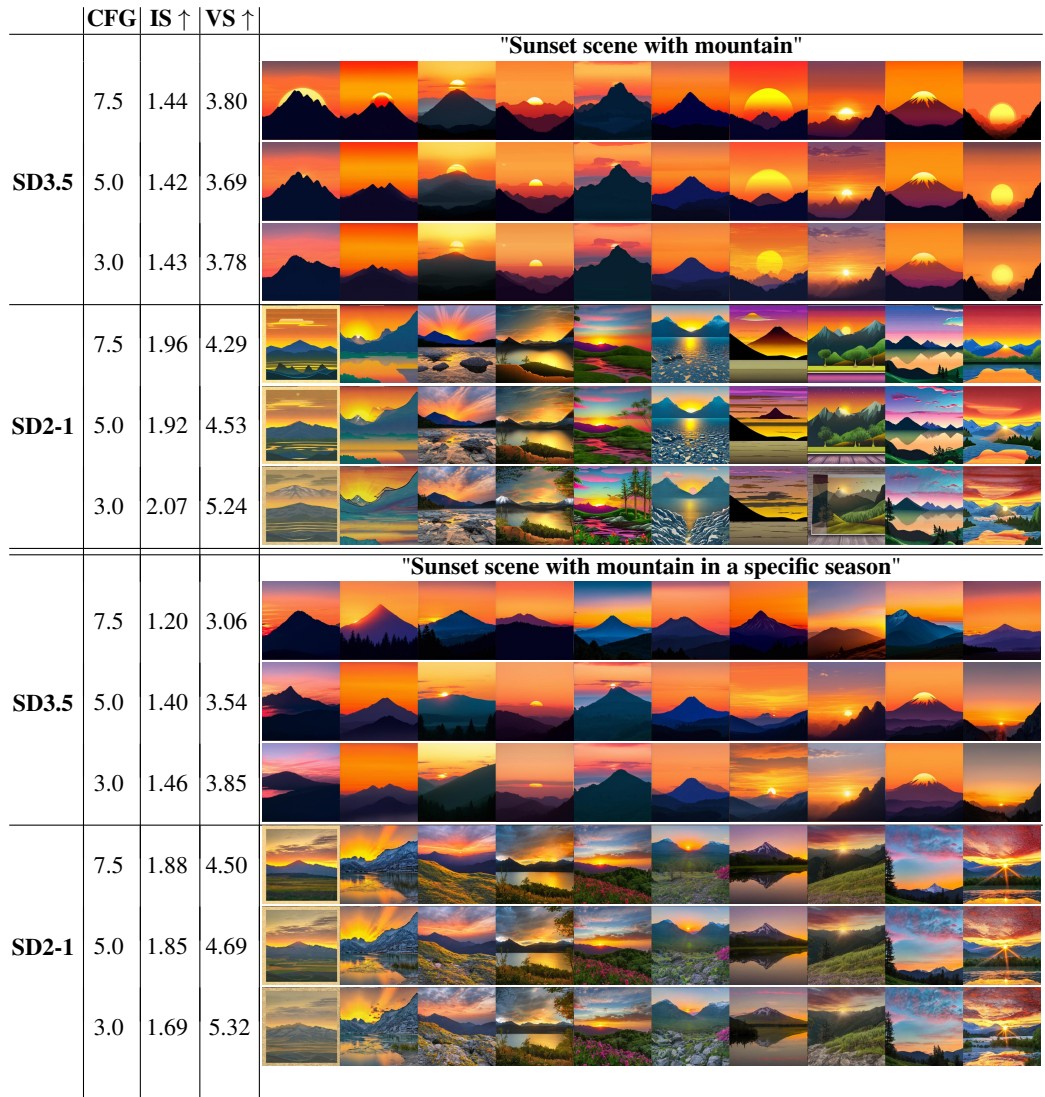

Figure 18: Comparison of the effect of classifier-free guidance (CFG) scale on SD baselines across different prompts. For each prompt, 40 images are generated to compute metrics, and 10 are displayed. Metrics include the Inception Score (IS) for image quality and the Inception Vendi Score (Vendi) for diversity assessment. **Across models, CFG scales, and prompts, there is no consistent pattern suggesting that reducing CFG yields more diverse images.** Specifically, we can observe that decreasing the CFG only affects the detail level of the objects (more realistic, sharper) without influencing the context or object choices. Therefore, reducing CFG does not improve diversity in context, which is the major strength of the proposed Rainbow.

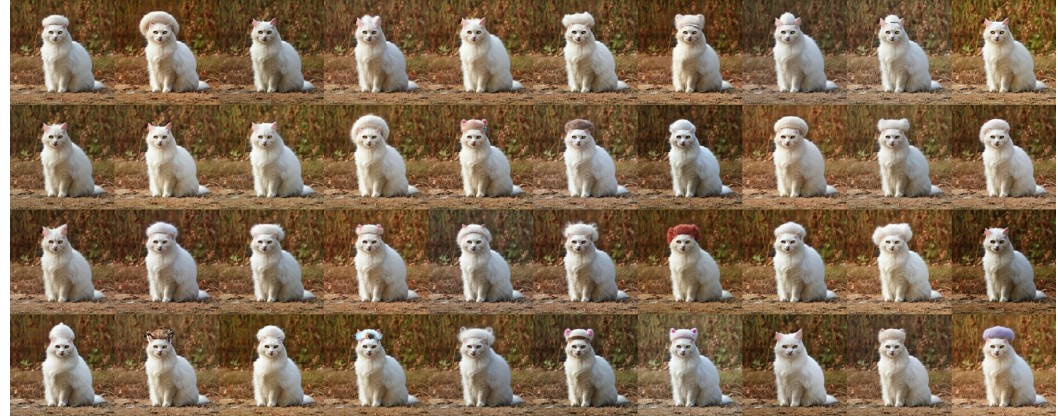

(a) DAC + SD2-1

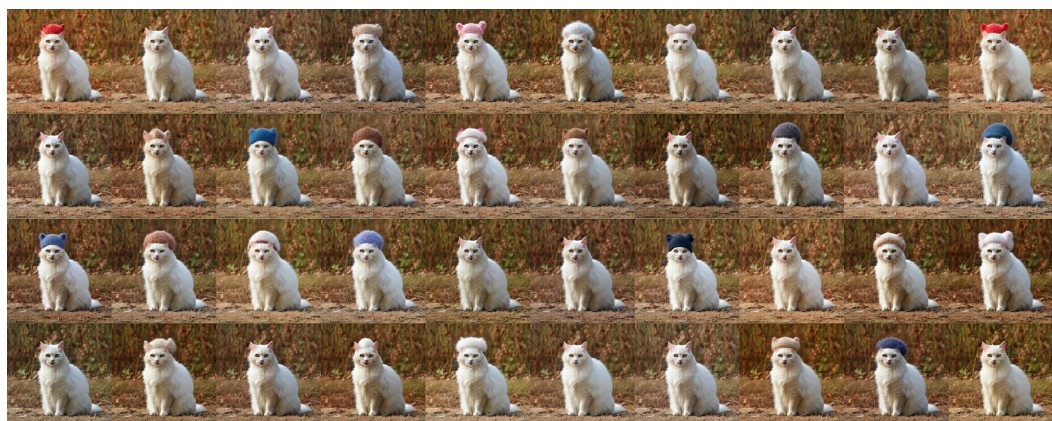

(b) DAC + Rainbow

Figure 19: Diversity in image-editing task. Given the original image of a cat (left) and the editing prompt "A cat wearing a wool cap", the baseline mostly produces white caps, while Rainbow captures diverse color choices of caps.

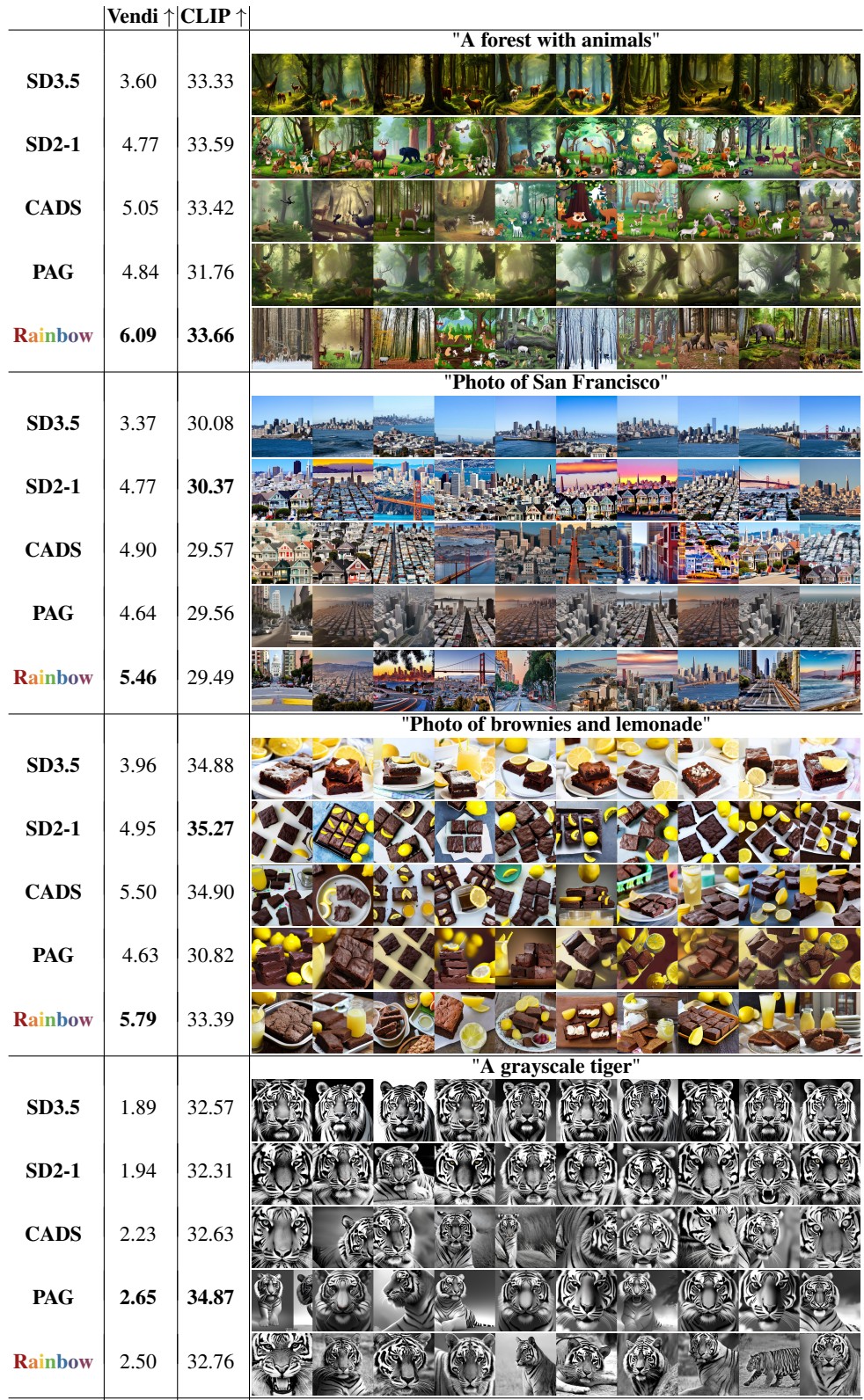

| | Vendi ↑ | CLIP ↑ |
|---|---|---|
| | | | "A forest with animals" |
| SD3.5 | 3.60 | 33.33 |
| SD2-1 | 4.77 | 33.59 |
| CADS | 5.05 | 33.42 |
| PAG | 4.84 | 31.76 |
| Rainbow | **6.09** | **33.66** |
| | | | "Photo of San Francisco" |
| SD3.5 | 3.37 | 30.08 |
| SD2-1 | 4.77 | **30.37** |
| CADS | 4.90 | 29.57 |
| PAG | 4.64 | 29.56 |
| Rainbow | **5.46** | 29.49 |
| | | | "Photo of brownies and lemonade" |
| SD3.5 | 3.96 | 34.88 |
| SD2-1 | 4.95 | **35.27** |
| CADS | 5.50 | 34.90 |
| PAG | 4.63 | 30.82 |
| Rainbow | **5.79** | 33.39 |
| | | | "A grayscale tiger" |
| SD3.5 | 1.89 | 32.57 |
| SD2-1 | 1.94 | 32.31 |
| CADS | 2.23 | 32.63 |
| PAG | **2.65** | **34.87** |
| Rainbow | 2.50 | 32.76 |

Figure 20: Diversity and quality across Rainbow and SD baselines. Each prompt generates 40 images per model. Notably, Rainbow captures seasonal variations in the "forest with animals" prompt and multiple perspectives of San Francisco beyond seas and buildings in SD baselines. For "brownies and lemonade", Rainbow generates diverse relevant objects on the table, and for "tiger", it provides multiple views, showcasing enhanced versatility and realism. The SD baselines tend to generate similar layouts and contexts across prompts, demonstrating less diversity.

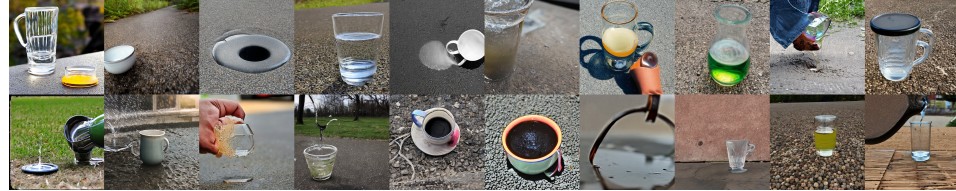

(a) Prompt: "A glass mug dropped to the ground"

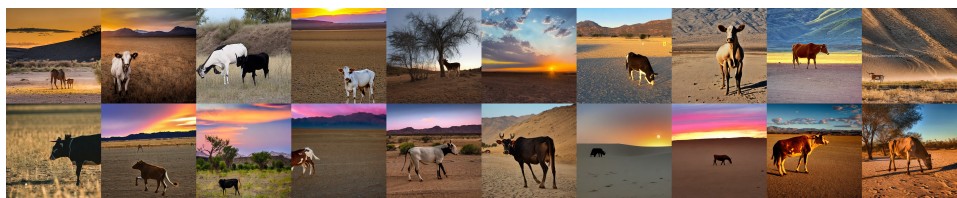

(b) Prompt: "A cow in a desert during a vibrant sunset"

Figure 21: Examples of cases that Rainbow perform not good.

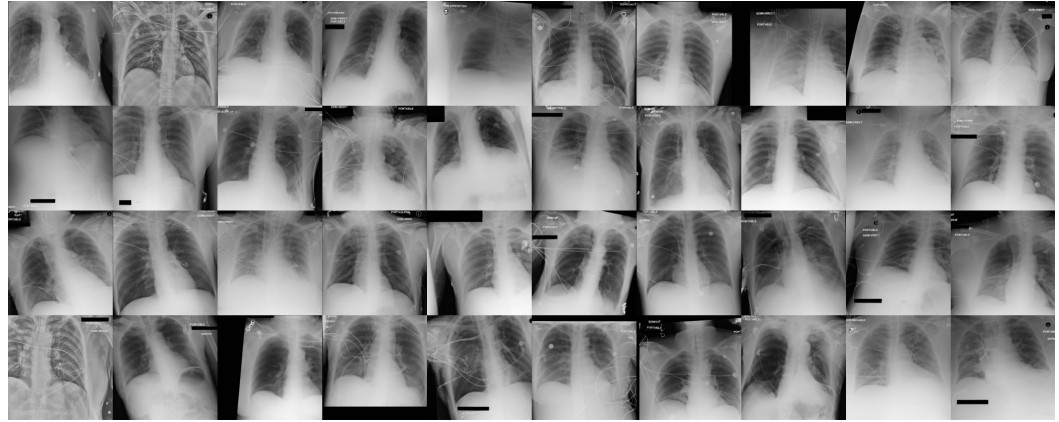

(a) Generations by RadEdit

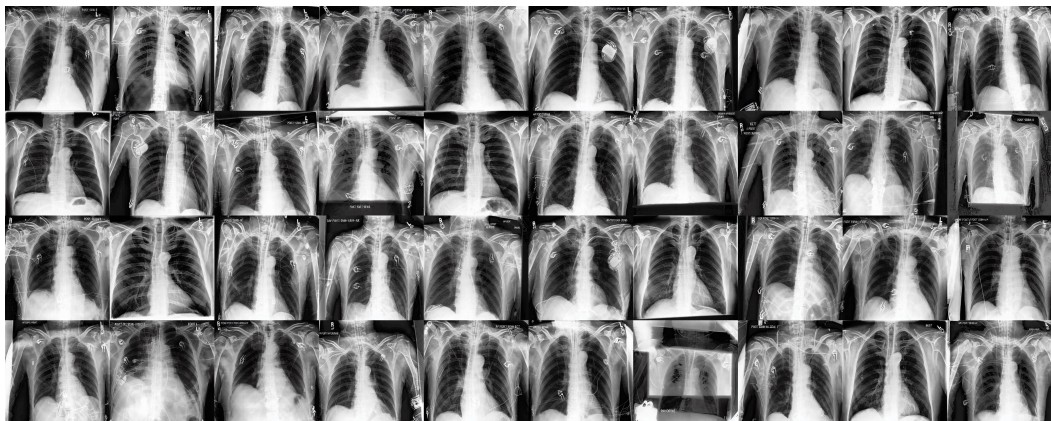

(b) Generations by SD1.5

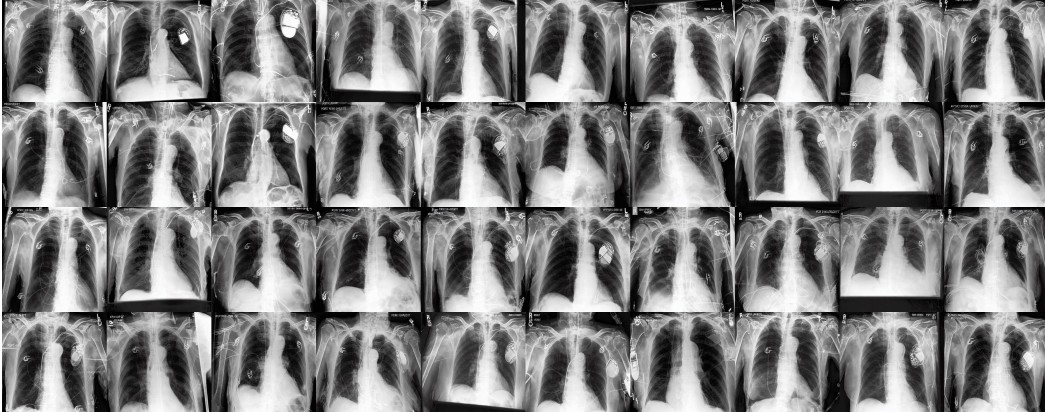

(c) Generations by Rainbow

Figure 22: Generations with the prompt "Chest X-ray showing support devices." Rainbow generates a more diverse set of medical devices compared to the SD model and RadEdit, while maintaining image quality comparable to the SD model.

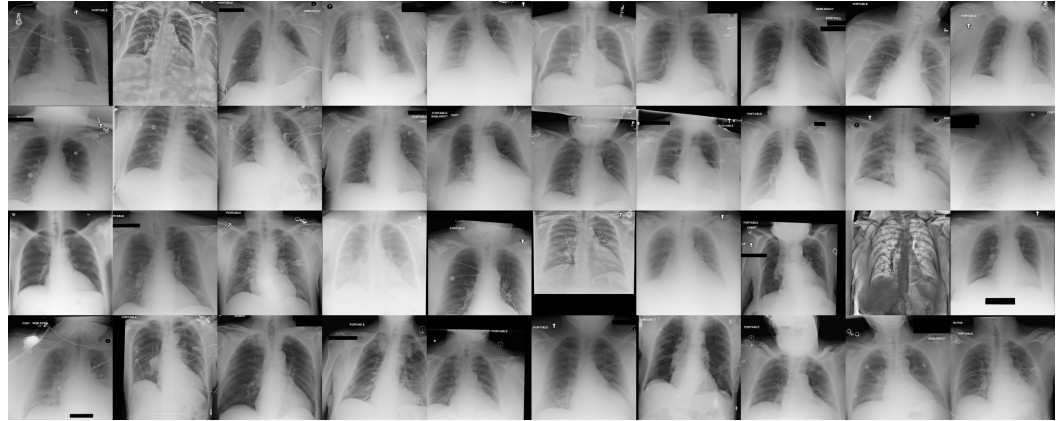

(a) Generations by RadEdit

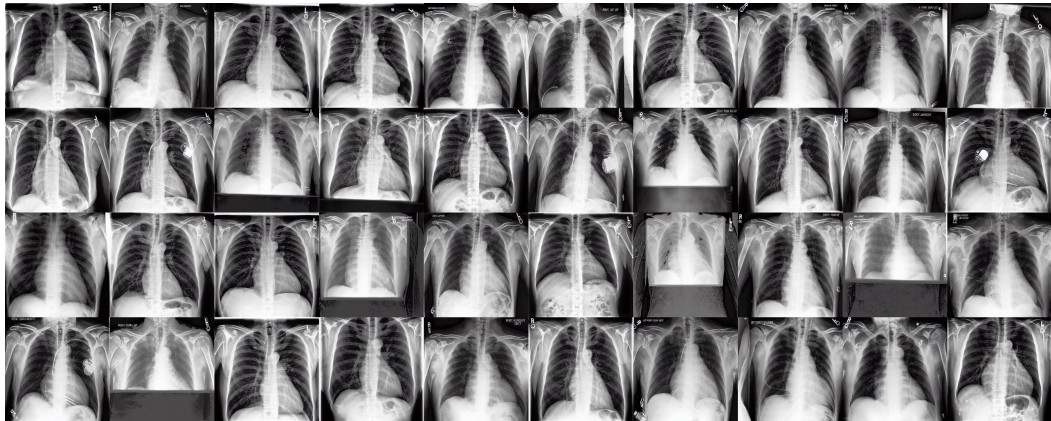

(b) Generations by SD1.5

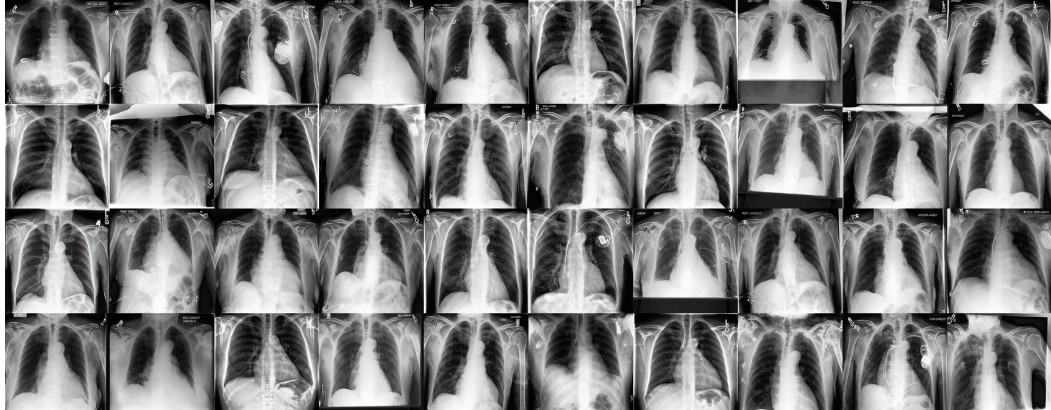

(c) Generations by Rainbow

Figure 23: Generations with the prompt "Chest X-ray showing Cardiomegaly". Rainbow shows more diversity in the anatomy of the generated chest X-ray, while SD mostly generates left and right lungs that are similar to each other. Furthermore, Rainbow provides diversity in the location of the generated support devices.

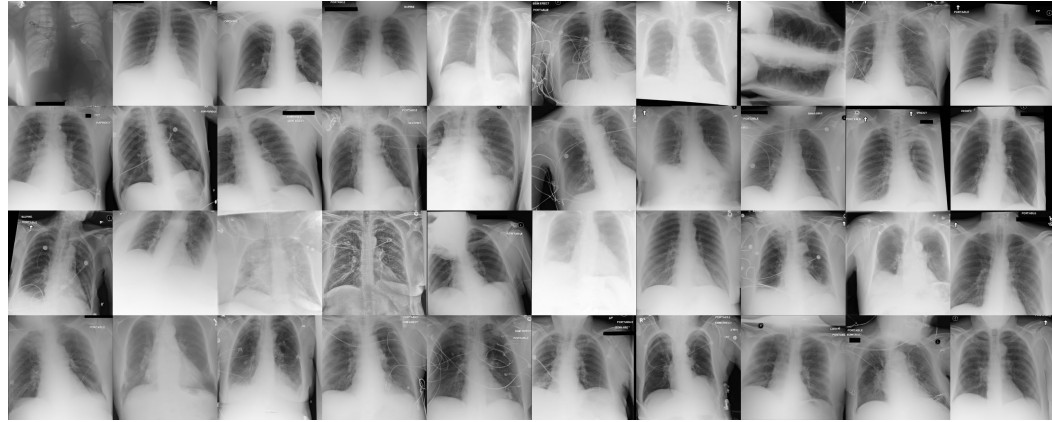

(a) Generations by RadEdit

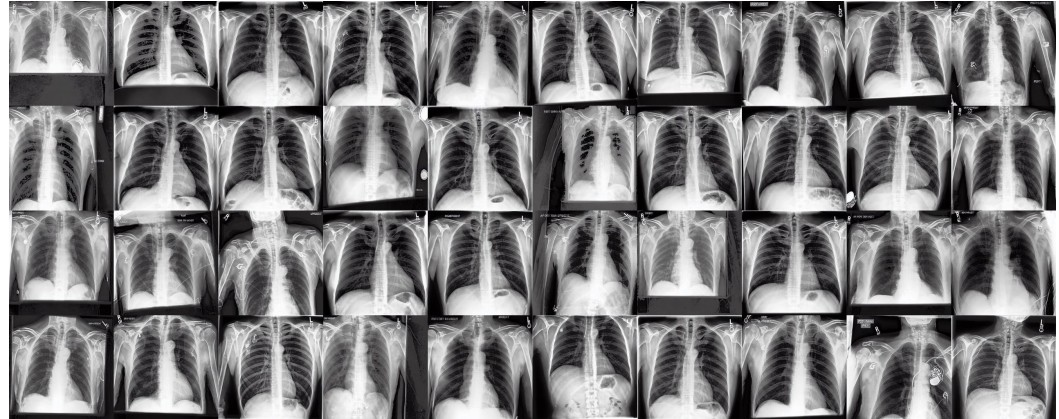

(b) Generations by SD1.5

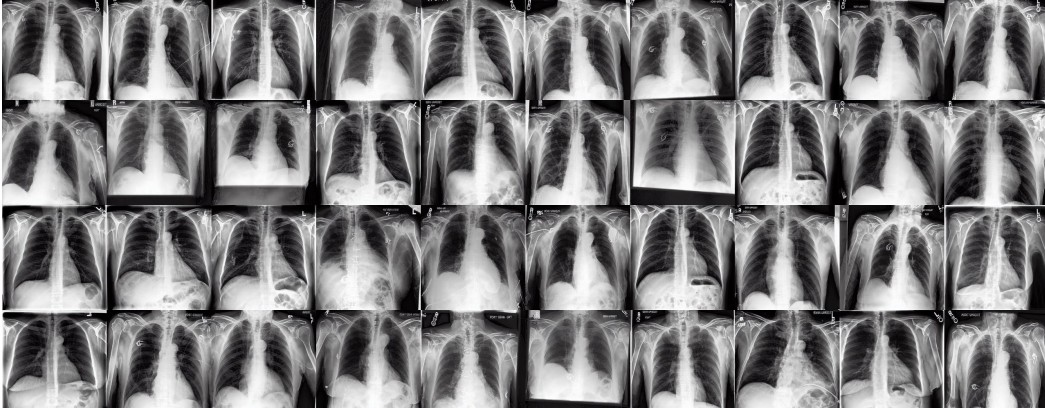

(c) Generations by Rainbow

Figure 24: Generations with the prompt "Chest X-ray with no significant findings". Rainbow shows more diversity in the anatomy of the generated chest X-ray, while SD has less variation in the anatomical structure of the lungs.

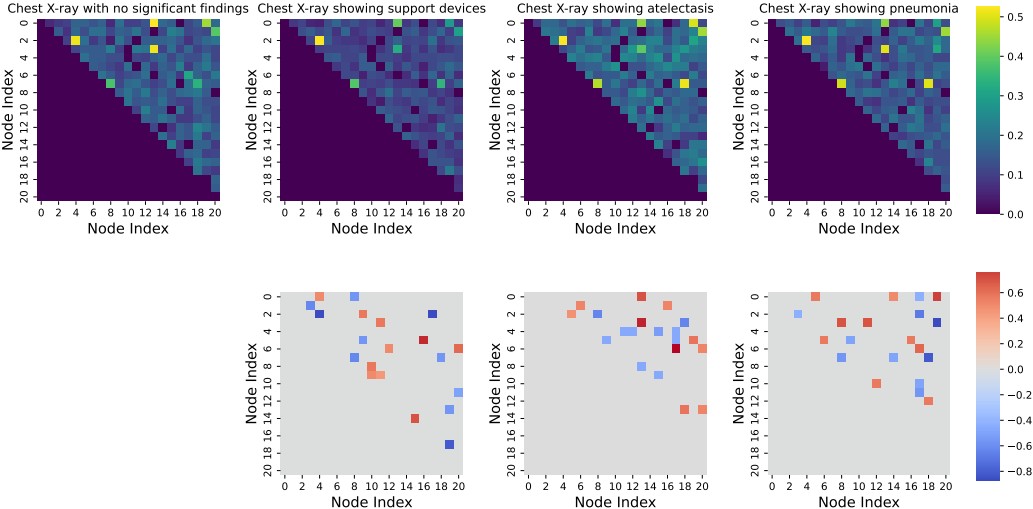

Figure 25: Heatmaps of edges across different prompts. In the first row, we present heatmaps of edges across 40 trajectories per prompt, with corresponding prompts labeled. In the bottom row, we illustrate the difference in edge distribution compared to the baseline prompt "Chest X-ray with no significant findings". We analyzed and extracted the 10 most frequently added edges and the 10 most frequently removed edges for the *difference heatmap*. Our observations reveal that (1) some edges consistently appear in most prompts with a significant proportion, and (2) it is evident that certain edges are representative of specific contexts. Particularly, in the difference heatmap, we can see certain edges are added with a high proportion, approximately 60%.

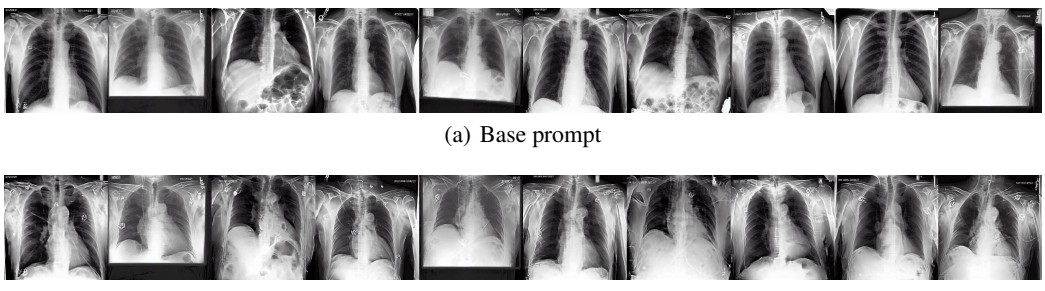

(a) Base prompt

(b) Base prompt + **suport devices** edges

Figure 26: Comparison of images generated by Rainbow with the base prompt "Chest X-ray with no specific findings" and with "support device" edges. By adding the edges corresponding to the entity "support devices" to the latent graph, we're able to modify the images with support devices. This demonstrates that Rainbow 's latent graphs encode structured and interpretable knowledge, and that manipulating these graphs enables fine-grained, concept-specific image editing.

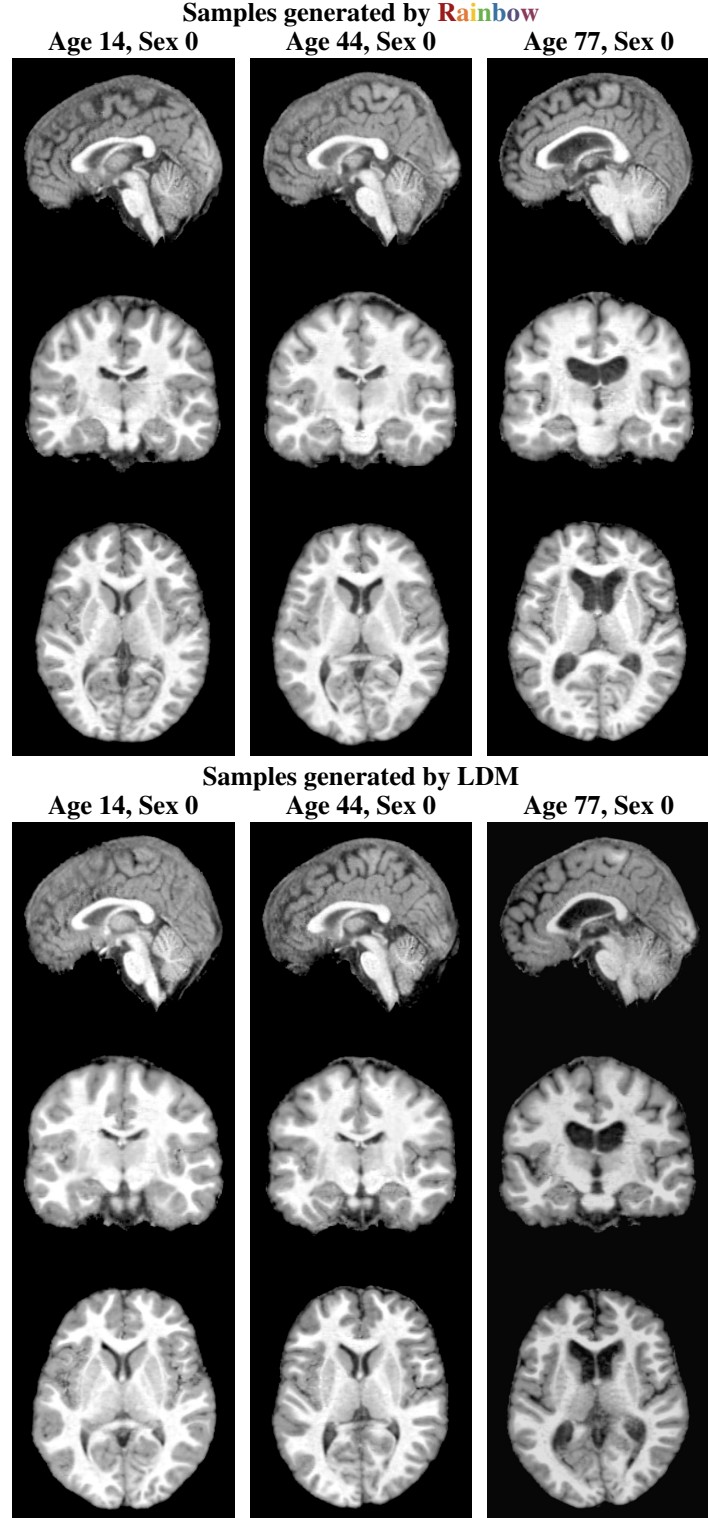

Figure 27: Brain MRI fidelity comparison

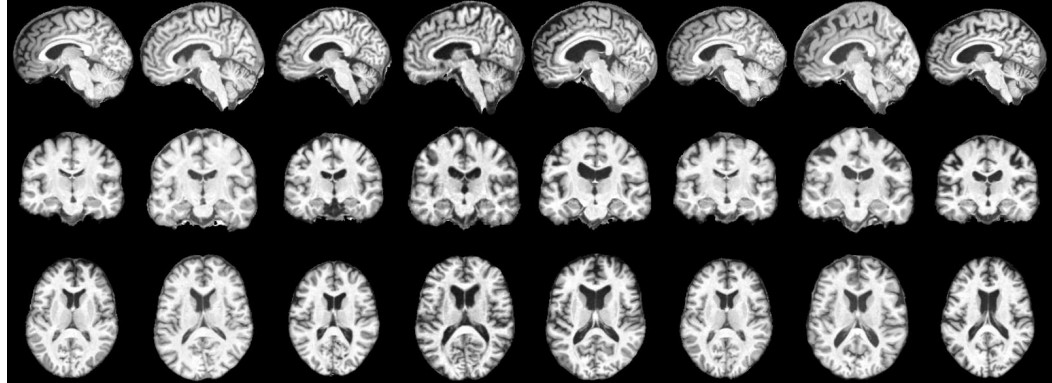

(a) Actual samples of male patients in age range from 60 to 65

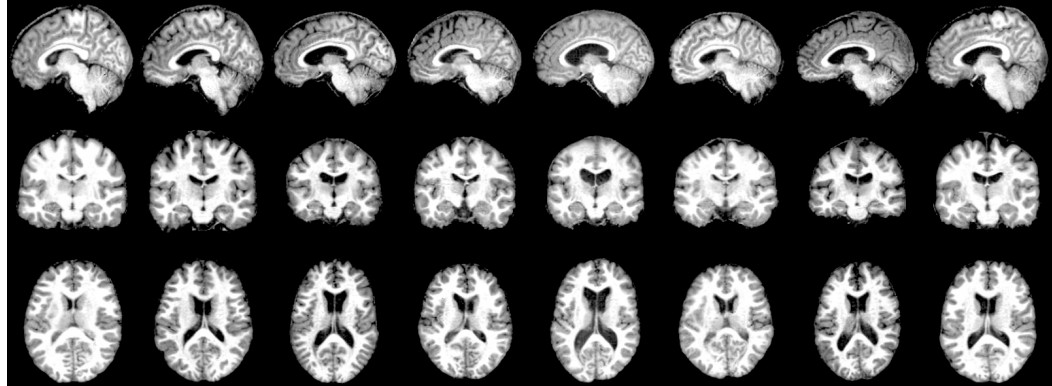

(b) Generated MRI by Rainbow

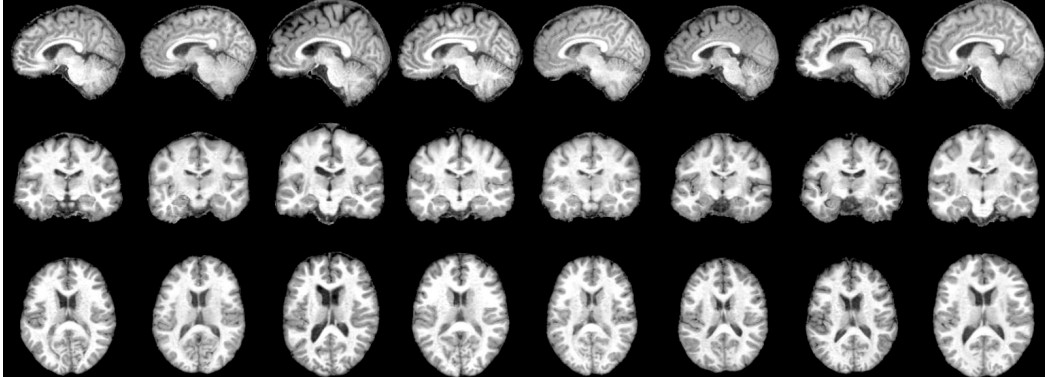

(c) Generated MRI by baseline LDM

Figure 28: Multiple generations per input condition comparison. The figure displays MRI images showcasing 8 samples generated from a single input condition. Given the input condition of a 65-year-old male, both Rainbow and the baseline LDM can generate plausible MRI images. However, compared to actual samples from males aged 60 to 65, it is evident that Rainbow captures a greater diversity in details, such as varying ventricle sizes. In contrast, the baseline LDM tends to generate images with consistently similar ventricle sizes, demonstrating less diversity.

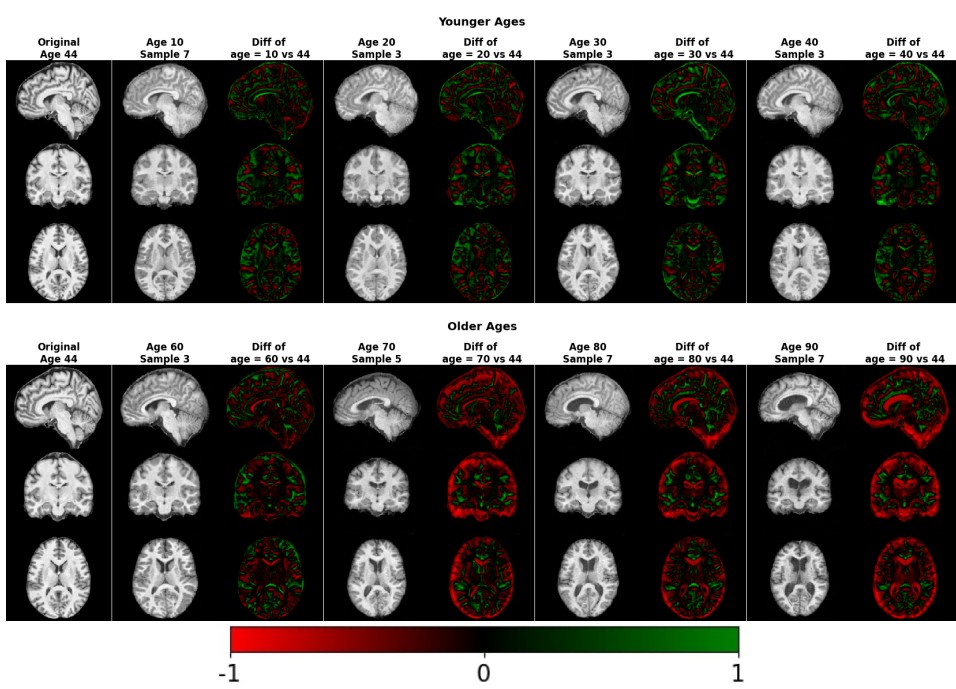

Figure 29: Counterfactual 3D Brain MRIs by Rainbow and the difference between the original image and generated counterfactuals

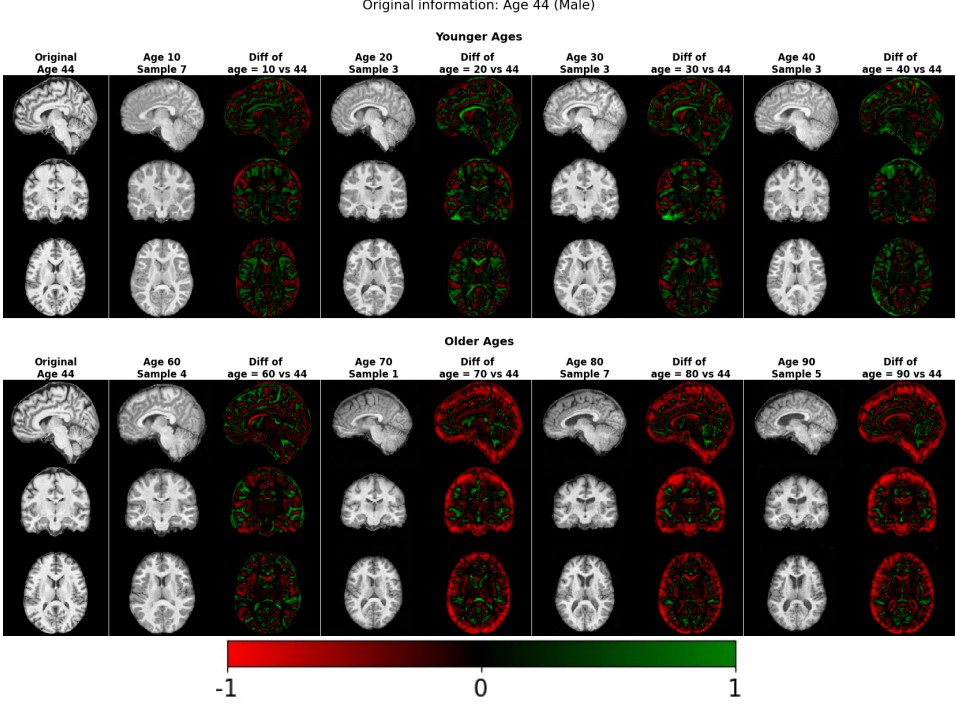

Figure 30: Counterfactual 3D Brain MRIs by baseline LDM and the difference between the original image and generated counterfactuals

