# OpenReview forum: "Discovering Latent Graphs with GFlowNets for Diverse Conditional Image Generation"
_NeurIPS.cc/2025/Conference — NeurIPS 2025 poster_

### Official Review · Reviewer_PWCV · 2025-06-30

**Clarity:** 3
**Significance:** 3
**Originality:** 3
**Rating:** 4
**Confidence:** 2

**Summary:**

Rainbow is a plug-and-play framework that adds explicit diversity to any pretrained conditional image generator. Instead of varying random seeds or paraphrasing prompts, Rainbow decomposes the input condition into multiple latent representations. Concretely, it inserts a latent graph—parameterised and sampled by a Generative Flow Network (GFlowNet)—into the prompt-encoding stage. Each GFlowNet trajectory captures a different plausible interpretation of the condition; feeding these trajectories through the frozen generator yields a batch of diverse, yet still faithful, images. Experiments on natural and medical datasets show simultaneous improvements in diversity (e.g., higher LPIPS / dispersion scores) and fidelity (lower FID) for image synthesis, conditional generation and counterfactual tasks, demonstrating Rainbow’s generality and effectiveness.

**Questions:**

How do graph depth, branching factor, and node-feature type affect both diversity and fidelity?
Could you provide an ablation that varies these hyper-parameters?

**Ethical Concerns:**

["NO or VERY MINOR ethics concerns only"]

**Final Justification:**

I would like to thank the author for the rebuttal points, i will increase my rating.

**Limitations:**

see weaknesses

**Quality:**

3

**Strengths And Weaknesses:**

Stenghs:
1. Rainbow can be bolted onto any pretrained conditional generator without retraining the base model, making it easy to adopt across diffusion, GAN and autoregressive backbones.
2.  Instead of random seeds or ad-hoc prompt edits, the method explicitly decomposes the condition into multiple latent routes, yielding diversity that is semantically tied to ambiguities in the original prompt.
3. Leveraging GFlowNets’ trajectory-sampling provides a mathematically grounded way to cover high-reward—but diverse—modes of the latent graph, avoiding mode collapse common in MCMC or pure random seeding.


Weaknesses:
1. Comparisons are mostly against “different seeds” or prompt-paraphrasing. State-of-the-art diversity techniques—e.g., DDIM stochasticity schedules, classifier-free guidance sweeps, Multimodal MCTS, or Latent Diffusion Ensembles—are missing, making it hard to gauge Rainbow’s real advantage.
2. Sampling diverse trajectories may lower prompt adherence or introduce artifacts. The paper reports FID and a single diversity metric but provides no human evaluation or CLIP-based alignment score to show that each variant still obeys the original condition.
3. GFlowNets are known to be sensitive to reward scaling and exploration temperature. The paper does not include convergence curves, variance across runs, or guidelines for hyper-parameter selection, which may hinder reproducibility.
4. The abstract states that each trajectory “captures an aspect of the uncertainty,” but no qualitative or quantitative analysis is given to link individual graph paths to human-interpretable factors (e.g., colour palette, viewpoint).

---

> ### Author Rebuttal · Authors · 2025-07-31
>
> 1.  **Comparisons are mostly against “different seeds” or prompt-paraphrasing. State-of-the-art diversity techniques—e.g., DDIM stochasticity schedules, classifier-free guidance sweeps, Multimodal MCTS, or Latent Diffusion Ensembles—are missing, making it hard to gauge Rainbow’s real advantage.**
>
> Thank you for your thoughtful review. We would like to take this opportunity to clarify our choices regarding baseline comparisons and to provide additional details about our methodologies.
>
> First, we would like to clarify that we do not solely rely on againsting “different seeds” or prompt-paraphrasing baselines; we also compared our method against baselines that intentionally trying to address the diversity of output images. Specifically, in our natural images experiment, we incorporated the following key baselines:
>
>     1.  **Traditional LDM:**  With this baseline, the "diversity" is not considered
>     2.  **CADS:**  With this baseline, the "diversity" is taken into account by employing a sampling strategy that anneals the conditioning signal. This is achieved by adding scheduled, monotonically decreasing Gaussian noise to the conditioning vector during inference.
>     3.  **PAG:**  This baseline considers "diversity" by not only incorporating prompt-paraphrasing but also by sampling from an unnormalized density function, enabling the generation of both high-quality and diverse prompts.
>
> In conclusion, our baselines cover a wide range of recent techniques aimed at addressing diversity effectively.
>
> Secondly, in Figure 18 in the Appendix, we also provide comparisons with variations in classifier-free guidance to highlight the benefit of Rainbow in generating diverse images in contextual level.
>
> Thirdly, it is worth noting that one of the novel ideas of Rainbow is to capture diversity by decomposing the condition into diverse representations. Consequently, our baselines include methods that generate a single output for uncertain input conditions, effectively highlighting the advantages of Rainbow in achieving greater diversity.
>
> Additionally, we include a comparison with the DDIM scheduler (implementation provided by stable-diffusion-webui) in the table below, demonstrating that Rainbow consistently outperforms baselines in both image quality and diversity, as well as ensuring obeying original prompt. About qualitative analysis, generations by DDIM have the same remaining limiation as other baselines, that is, images are non-identical but similar in context, not capture the diverse representation of the input prompt's uncertainty. For instance, generation with "Sunset scene with mountain in a specific season" produce almost all spring season over 40 images. We will add this additional analysis to the revised version of the manuscript..
>
>
> | **Model name**    | **IS ↑** | **CLIP ↑**  | **Pixcel VS ↑**  | **Inception VS ↑**  |
> |-------------------|----------|-------------|------------------|---------------------|
> | **SD3.5**         | 7.37     | 30.29       | 3.75             | 25.09               |
> | **SD2-1**         | 8.49     | 30.27       | 3.74             | 25.63               |
> | **CADS**          | 9.46     | 30.27       | 3.88             | 27.85               |
> | **PAG**           | 9.93     | 30.21       | 3.90             | 26.92               |
> | **DDIM**          | 8.44     | 30.28       | 3.91             | 26.62               |
> | **Rainbow**       | **10.45**| **30.32**   | **3.94**         | **28.90**           |
>
>
> Finally, we apologize for not being able to find any studies on MCTS or Latent Diffusion Ensembles specifically applied to image generation tasks. If you have any references or further details regarding these approaches, it would be greatly appreciated if you could share them, as we are eager to explore relevant techniques to enhance our research.
>
>
>
>
> ---------------
>
> 2.  **Sampling diverse trajectories may lower prompt adherence or introduce artifacts. The paper reports FID and a single diversity metric but provides no human evaluation or CLIP-based alignment score to show that each variant still obeys the original condition.**
>
> Thank you for your insightful feedback.
>
> **To address your concerns regarding obeying original condition**, we would like to clarify that we have provided evaluations based on the CLIP score for experiments with prompt-based conditions. Specifically:
>
> -   **For natural images**, we present the CLIP score in line 249, indicating that Rainbow achieves significantly higher diversity while preserving or slightly increasing the CLIP score at around 30.3. This result demonstrates the method's ability to maintain prompt adherence. Further details can be found in Table 3 in the appendix (we also copied this table and present it in the answer for your first question above).
>
> -   **For chest X-ray images**, we include a CLIP evaluation in line 273, where we report that all models achieve similar CLIP scores at 33.5. This indicates that Rainbow improves diversity over the baseline while still adhering to the original prompts. The full CLIP scores are provided in Table 4 in the appendix.
>
>
> **To address your concerns regarding human evaluation on adherence to original conditions**, we present the results of our human evaluation on the generated chest X-rays. In our study, we conducted human evaluations using 8 prompts, with 12 images generated per prompt. Three evaluators, all of whom are medical doctors, assessed the realism of the images by selecting which model—either the baseline SD or Rainbow—exhibited higher realism based on the given prompt descriptions. The results showed that **75% of the evaluators voted in favor of the Rainbow model**.
>
> We are expanding the human evaluation across all three datasets in our future work, along with increasing the number of evaluators / experts involved.
>
>
>
> We hope this clarifies how our approach balances diversity and prompt adherence.
>
>
>
>
>
> -----------
>
> 3.  **GFlowNets are known to be sensitive to reward scaling and exploration temperature. The paper does not include convergence curves, variance across runs, or guidelines for hyper-parameter selection, which may hinder reproducibility.**
>
> Thank you for your insightful comment.
>
> Rainbow is built on a pretrained image encoder-decoder and U-Net models, contributing to its stability and reducing sensitivity during GFlowNet training. We will ensure to include reward and loss curves in the final version of the paper to enhance the clarity of our findings.
>
> **Regarding hyperparameter selection for GFlowNet**, specifically for the parameters M, N, and ρ, our choices were based on several factors, including computational resources, the need for diversity, task plausibility, and results from fine-tuning.
> We adjusted the value of N by using a higher value for natural images to allow greater freedom in diversity and concept choices, while for medical datasets, we opted for a lower N value to balance diversity and ensure that the images still follow a domain-specific template. We maintained a relatively high graph sparsity, above 0.6; higher sparsity results in fewer sampled edges. This strategy enabled us to maximize the number of graphs given our available computational resources.
> We fine-tuned the combinations of M, N, and ρ for optimal performance in terms of both diversity and image quality. Notably, we observed that all combinations produced similar CLIP scores for prompt delivery evaluation. The fine-tuning results for chest X-rays are provided in Figure 11 in the appendix (referred to from line 313).
>
> We hope this addresses your concerns.
>
> ---------------
> 4.  **The abstract states that each trajectory “captures an aspect of the uncertainty,” but no qualitative or quantitative analysis is given to link individual graph paths to human-interpretable factors (e.g., colour palette, viewpoint).**
>
> We appreciate your observation and acknowledge the importance of linking graph trajectories to interpretable features. We would like to clarify that Rainbow demonstrating clear benefits in improving diversity, both quantitatively and qualitatively, our analysis does show that the graph can be interpreted to human-understandable features such as "seasons" and "support devices" in Section 4.2.2. This supports our claim that each trajectory "captures an aspect of the uncertainty".
>
> In addition, we agree that we need one step further to establish a more direct connection between individual graph paths and specific interpretable factors. We plan to address this direct interpretability in the next paper for a big upgraded version of Rainbow as part of our future research.
>
> We hope this answer clarifies your concerns!
>
>
> -----------
> 5. **How do graph depth, branching factor, and node-feature type affect both diversity and fidelity? Could you provide an ablation that varies these hyper-parameters?**
>
> Reffering to the fine-tuning results presented in Figure 11 in the appendix, we observe the effects of the number of nodes (N), sparsity ρ that affect the number of edges on sampled trajectories, and the number of trajectories (M) on diversity (as measured by the VS score) and fidelity (as measured by the FID score) as follows:
>
> -   Configurations with M=5 and a sparsity level of ρ=0.64, regardless of the value of N, tend to produce lower scores in diversity and exhibit variability in image quality.
> -   Configurations with N=20 and ρ=0.82 (a mid-range value), regardless of the value of M, tend to stabilize image quality while also generating highly diverse outputs.
> -   Sparsity levels of ρ=0.88, independent of the other configurations, result in instability, producing both low and high scores in terms of diversity and quality.
>
> Based on these fine-tuning results, we selected the combination that achieves a balance between image quality and diversity by choosing the configuration with the highest ratio of VS^3/FID, which is N\=20, M\=10, and ρ\=0.82.
>
> We hope this answer clarifies your question.

---

> ### Comment · Reviewer_PWCV · 2025-08-01
> **Comments**
>
> Thanks for your rebuttal.
> The authors addressed my comments.

---

> ### Author Response · Authors · 2025-08-01
>
> Thank you for reviewing the rebuttal, and we appreciate your prompt reply. As your concerns have been addressed, would you consider raising the evaluation score?
>
> If you still have concerns, please let us know. We are available to address any additional questions you may have.
>
> Thank you again for your constructive feedback.

---

> > ### Comment · Reviewer_PWCV · 2025-08-05
> >
> > I will raise my score from 3 to 4

---

> > > ### Author Response · Authors · 2025-08-06
> > >
> > > Thank you for your time and effort in reviewing my submission and rebuttal, as well as for increasing the score; we truly appreciate your recognition and support!

---

### Official Review · Reviewer_Jz2s · 2025-06-30

**Clarity:** 3
**Significance:** 4
**Originality:** 3
**Rating:** 5
**Confidence:** 4

**Summary:**

This paper introduces "Rainbow," a novel framework designed to enhance the diversity of conditional image generation models. The core problem addressed is the inherent ambiguity in input conditions (e.g., text prompts), which can correspond to multiple plausible image outputs. The authors' key idea is to decompose a single input condition into multiple, diverse latent representations. This is achieved by introducing a latent graph structure and employing Generative Flow Networks (GFlowNets) to sample a set of diverse trajectories (subgraphs) over this graph. The method's efficacy is demonstrated across multiple domains, including natural image generation (Flickr30k), 3D medical imaging (Brain MRIs), and 2D medical imaging (Chest X-rays). The results show consistent improvements in both diversity (measured by Vendi Score) and image quality (measured by IS and FID) compared to strong baselines like various versions of Stable Diffusion, CADS, and PAG. The paper also provides compelling evidence for the interpretability of the learned latent graphs and the positive impact of the generated data on downstream tasks.

**Questions:**

1. Regarding the latent graph interpretation (Section 4.2.2): How were the sets of edges corresponding to specific concepts (e.g., "spring edges") identified? Was this an automated clustering process based on edge frequency, or was there manual selection involved after clustering the output images? Please clarify the procedure.

2. Could you elaborate on the choice of the RNN in the graph decoder ($Q_D$)? Since the final graph is an unordered set of edges, is the sequential processing of the RNN critical? Have you experimented with non-sequential architectures like a simple pooling or a Transformer encoder?

3. In Table 1, the performance of the age/sex classifiers on real data is provided as a reference. What is the performance of these classifiers on the training set of the real data? This would help contextualize the generalization gap and better interpret the results on the synthetic data.

**Ethical Concerns:**

["NO or VERY MINOR ethics concerns only"]

**Limitations:**

yes

**Quality:**

4

**Strengths And Weaknesses:**

Strengths

Novel and Sound Methodology: The primary strength of this work lies in its innovative integration of GFlowNets with latent diffusion models. Using GFlowNets to explicitly model a distribution over diverse graph structures that represent conditional uncertainty is a principled and powerful approach. It directly tackles the mode-seeking behavior of many generative models and provides a more structured way to explore the solution space than simple noise manipulation. The technical execution, including the use of the Detailed Balance objective and the design of the graph generator and decoder, is well-reasoned and clearly explained.

Comprehensive Empirical Evaluation: The authors have conducted an extensive set of experiments across three distinct and challenging domains. The visual comparisons (e.g., Figure 2, "Sunset scene with mountain") are convincing. Rainbow clearly generates a wider variety of seasons, lighting, and compositions compared to the baselines, which often produce repetitive or stylistically limited outputs. The quantitative analysis (Figure 4, Table 1, Table 3) supports the claims. Rainbow consistently achieves a better trade-off between diversity (higher Vendi Score) and fidelity (higher IS, lower FID) than the baselines.

Model-Agnostic and Flexible Framework: An advantage of Rainbow is its applicability to any pretrained conditional generative model. By operating on the condition representation and keeping the core generative model frozen, it serves as a versatile plug-in. This is impressively demonstrated by its successful application to different Stable Diffusion versions for natural images and chest X-rays, and a MONAI-optimized LDM for 3D brain MRIs.


Weaknesses

Computational Overhead: The paper acknowledges that the method requires higher computational resources due to the parallel generation of M trajectories (graphs). While the results justify the cost, a more detailed analysis of the trade-off between the number of graphs (M) and performance versus computational cost would be beneficial. The ablation study (Figure 10) shows that performance degrades as M decreases, but the cost-benefit analysis isn't fully fleshed out.

Clarity on Graph Hyperparameters: The choice of the latent graph size (N nodes) and sparsity (ρ) seems somewhat arbitrary. The ablation study explores these parameters for the Chest X-ray experiment, but a more in-depth discussion on the intuition behind selecting these values for different domains would strengthen the paper. For instance, why was N=20 chosen for natural images but N=8 for brain MRIs? Is there a principled way to set these, or is it purely empirical?

Limited Discussion on Failure Cases: While the presented results are strong, the paper would be more complete with a discussion of failure modes. Are there types of conditions or prompts where Rainbow struggles to generate diversity or maintain fidelity? For example, how does it handle highly complex prompts with many entangled concepts?

Difference with Subspace-based Methods: It seems that the core goals of rainbow and subspace-based representation learning methods are very similar. Why not use subspace methods to learn different semantic representations?

---

> ### Author Rebuttal · Authors · 2025-07-31
>
> 1.  **Regarding the latent graph interpretation (Section 4.2.2): How were the sets of edges corresponding to specific concepts (e.g., "spring edges") identified? Was this an automated clustering process based on edge frequency, or was there manual selection involved after clustering the output images? Please clarify the procedure.**
>
>
> Thank you for your question. The process is automatic. We rank the top 10 added edges when the prompt changes from _"sunset scene with mountain"_ to _"sunset scene with mountain in a specific season"_, and then append these edges to the edge set of the first prompt.
>
> A similar procedure was used in the chest X-ray experiment shown in Figure 7, with the corresponding edge heat map provided in Figure 24 (referred to in line 287).
>
> Hope this answers your question.
>
>
>
>
> -----------------
> 2.  **Could you elaborate on the choice of the RNN in the graph decoder ( QD)? Since the final graph is an unordered set of edges, is the sequential processing of the RNN critical? Have you experimented with non-sequential architectures like a simple pooling or a Transformer encoder?**
>
> First, we would like to clarify that our edge sampling process is **order-dependent**—we sample edges **one at a time**, and the choice of each edge can influence the selection of subsequent ones. Therefore, maintaining the order of edges is important, and a sequential model like an RNN helps capture this dependency during decoding.
>
> Second, we would like to mention that our current design **does include a pooling layer after the RNN**, allowing the model to aggregate information across all sequentially sampled edges before making the final predictions.
>
> Third, we conducted an ablation study comparing Rainbow with RNN (RNN -> Pooling) and without the RNN (directly to Pooling) in the graph decoder in the brain MRI experiment. The results are shown below:
>
> | Method | Synthesized Sample (Age MAE)  ↓| Counterfactual on Age (Age MAE) ↓ | Counterfactual on Sex (Age MAE) ↓ |
> | --- | --- | --- | --- |
> | Rainbow w RNN | **19.87** | **15.40** | **14.36** |
> | Rainbow w/o RNN | 23.91 | 20.42 | 19.71 |
>
> We can see that removing the RNN leads to a noticeable drop in performance across all evaluation settings. This suggests that modeling sequential dependencies among sampled edges is beneficial for generating more accurate and consistent outputs.
>
>
>
> ------------------
> 3.  **In Table 1, the performance of the age/sex classifiers on real data is provided as a reference. What is the performance of these classifiers on the training set of the real data? This would help contextualize the generalization gap and better interpret the results on the synthetic data.**
>
> Thank you for your question.
>
> We’d like to provide more context on the "Real data" results in Table 1. We sampled MRIs and corresponding conditions from the real dataset to match the age and sex distribution of the generated samples shown in the bottom rows of the table. This ensures a fair comparison when evaluating classifier performance.
>
>
>
>
>
>
> ------------------
> 4. **Computational Overhead: The paper acknowledges that the method requires higher computational resources due to the parallel generation of M trajectories (graphs). While the results justify the cost, a more detailed analysis of the trade-off between the number of graphs (M) and performance versus computational cost would be beneficial. The ablation study (Figure 10) shows that performance degrades as M decreases, but the cost-benefit analysis isn't fully fleshed out.**
>
> Thank you for your question.
>
> We would like to clarify that across all datasets in our experiments, varying the number of trajectories does not significantly affect the total generation time, as all the trajectories are predicted as output from the GFlownet at once. This parallel processing enables efficient inference, even when generating a larger number of trajectories, without introducing substantial delays.
>
>
> For more detail, we did an analysis on natural image generation, we measured inference time for Rainbow and SD2-1 when generating 40 images per prompt (averaged over 60 prompts). The average time per image for Rainbow is approximately 0.40 seconds, which is comparable to SD2-1 at 0.39 seconds. The forward pass of the Rainbow module—which decomposes the original prompt into 40 prompt representations—takes only 0.104 seconds per prompt. This forward duration is similar when we generate 40, 20, and 10 graphs.
>
> Hope this clarify your concerns.
>
> ---------------
> 5. **It seems that the core goals of rainbow and subspace-based representation learning methods are very similar. Why not use subspace methods to learn different semantic representations?**
>
> Thank you for the insightful suggestion.
>
> Rainbow is our initial attempt at decomposing a single condition into multiple structured representations. We see it as a first step that opens up promising directions for future research and improvement.
>
> We agree that subspace-based approaches are meaningful and can offer benefits, particularly in terms of interpretability. However, training with predefined subspaces can introduce several challenges—for example, determining how to cluster data into subspaces, deciding the number of subspaces, and defining their semantic scope. In contrast, Rainbow offers a more flexible framework by allowing these representations to emerge naturally during training and enabling interpretability through post-hoc analysis of the learned graphs.
>
> We see both approaches as complementary and believe exploring their integration could be a fruitful avenue in future work.
>
> ---
>
>
>
>
> 6. **why was N=20 chosen for natural images but N=8 for brain MRIs? Is there a principled way to set these, or is it purely empirical?**
>
> Thank you for your question.
>
> **Regarding hyperparameter selection for GFlowNet**, specifically for the parameters M, N, and ρ, our choices were based on several factors, including computational resources, the need for diversity, task plausibility, and results from fine-tuning.
> We adjusted the value of N by using a higher value for natural images to allow greater freedom in diversity and concept choices, while for medical datasets, we opted for a lower N value to balance diversity and ensure that the images still follow a domain-specific template. We maintained a relatively high graph sparsity, above 0.6; higher sparsity results in fewer sampled edges. This strategy enabled us to maximize the number of graphs given our available computational resources.
> We fine-tuned the combinations of M, N, and ρ for optimal performance in terms of both diversity and image quality. Notably, we observed that all combinations produced similar CLIP scores for prompt delivery evaluation. The fine-tuning results for chest X-rays are provided in Figure 11 in the appendix (referred to in line 313) for reference.
>
> We hope this answer clarifies your question.
>
>
> --------------
> 7. **Are there types of conditions or prompts where Rainbow struggles to generate diversity or maintain fidelity? For example, how does it handle highly complex prompts with many entangled concepts?**
>
> Thank you for your question.
>
> First, we would like to note that even very detailed prompts, such as _"A serene forest scene at sunrise, with golden rays filtering through the trees, vibrant wildflowers in the foreground, and a gentle stream flowing, reflecting the colors of the morning light."_, often still contain uncertainty in object types and layout choices. This suggests that capturing diversity remains valuable, as it can be challenging to express every visual detail purely through text.
>
> Second, we have observed some cases where both the baselines and Rainbow struggle to accurately reflect certain prompt details. For example:
>
> -   _"A glass falls to the ground"_ → models may miss the causal implication (e.g., the glass breaking).
> -   _"#num of cats standing evenly horizontally on the street"_ → models often struggle to generate the exact number of cats and align them evenly.
>
> It is worth noting that images that are unable to convey specific implications can also stem from limitations inherent to the diffusion model itself. We appreciate this important observation and will include some of these failure cases in the revised version of the paper.

---

> > ### Comment · Reviewer_Jz2s · 2025-08-07
> >
> > Thank you for your rebuttal, which has addressed my concerns.

---

> > > ### Author Response · Authors · 2025-08-07
> > >
> > > Thank you for your time and effort in reviewing my submission and rebuttal, as well as for maintaining your positive evaluation; we truly appreciate your support and feedback!

---

### Official Review · Reviewer_75uy · 2025-07-02

**Clarity:** 2
**Significance:** 3
**Originality:** 4
**Rating:** 5
**Confidence:** 2

**Summary:**

Rainbow is a new method for generating diverse images from text prompts or other conditions. The main idea is that when given a prompt like "sunset scene with mountain" there are many valid ways to interpret it - different seasons, lighting, perspectives, etc. Current methods struggle to capture this diversity, which was shown in different works. Rainbow addresses this by using Generative Flow Networks (GFlowNets) to create multiple graph trajectories that represent different interpretations of the input condition. Each trajectory gets decoded into a different latent representation, which then generates different image from the trajectory.
The authors test this on three datasets: natural images, brain MRIs, and chest X-rays. Results show Rainbow generates more diverse images while maintaining quality compared to baselines like Stable Diffusion.
The key innovation is connecting GFlowNets (which are good at sampling diverse solutions) with diffusion models (which generate high-quality images). The learned graphs also seem interpretable - certain edges correspond to concepts like "winter" or "devices."

**Questions:**

1. How did you decide on using graph structures to represent uncertainty? Were other structured representations considered or related works can be improved for that then?
2. What is the computational overhead of Rainbow compared to standard diffusion models - can you add this to the paper to make it clear? How does inference time scale with the number of graphs M?
3. When given a very specific prompt with little ambiguity (like a red circle on white background), does Rainbow still try to force diversity?  How does it handle low-uncertainty conditions?
4. Are the interpretable edges (e.g. seasons) consistent across different training runs or did you select the best examples?
5. Is there a principled way to choose M, N, and $\rho$ for a new domain, or does it require extensive hyperparameter search -- not sure I feel the intuition here?

**Ethical Concerns:**

["NO or VERY MINOR ethics concerns only"]

**Final Justification:**

Authors produced nice results with innovative approach and addressed all issues during rebuttal, I would like to save my positive score given all discussions.

**Limitations:**

The method requires training additional components top of pre-trained models, which adds complexity.
No principled way to set hyperparameters - each domain uses very different values
The approach may not scale well to high-resolution images due to memory requirements of maintaining M parallel trajectories
The interpretability of edges is only demonstrated through cherry-picked examples; no systematic evaluation of interpretation stability
Assumes all input conditions have meaningful uncertainty to exploit, which may not always be true
The connection between graph sparsity and semantic diversity is assumed rather than proven

**Paper Formatting Concerns:**

no concerns

**Quality:**

4

**Strengths And Weaknesses:**

### Strengths:
- Novel combination of GFlowNets with diffusion models that hasn't been explored before
- Strong experimental results across three very different domains (natural images, brain MRI)
- The learned graph structures show some interpretability - can identify edges corresponding to seasons or medical devices
- Comprehensive evaluation with multiple metrics (IS, VS, FID, CLIP scores)
- Works with any pre-trained conditional diffusion model without major modifications

### Weaknesses:

- Computational cost seems high (M=40 parallel trajectories) but no timing comparisons provided
- Hyper parameter selection (nodes, graphs, sparsity) appears arbitrary and domain-specific
- The method might be forcing diversity even when the prompt is very specific
- Unclear how stable the interpretable edges are across different training runs
– minor thing  hard to distinguish (trajectories vs graphs) as notation

---

> ### Author Rebuttal · Authors · 2025-07-31
>
> 1.  **How did you decide on using graph structures to represent uncertainty? Were other structured representations considered or related works can be improved for that then?**
>
> Graph structure was the only representation we considered. Driven by the objective of generating multiple output images while capturing a shared context, GFlowNet was chosen early on and GFlowNet primarily operates on graph structures.
>
> The key design choice was **how to sample the graph**. If we add one **node at a time**, the resulting graph resembles a **causal graph**, where edges tend to connect all nodes. In contrast, adding one **edge at a time** yields a structure more like a **knowledge graph**, potentially with multiple disconnected subgraphs.
>
> We chose the **edge-by-edge** sampling strategy and found that it performed better in practice.
>
> Hope this answers your question.
>
> ------------
> 2.  **What is the computational overhead of Rainbow compared to standard diffusion models - can you add this to the paper to make it clear? How does inference time scale with the number of graphs M?**
>
> Thank you forthe suggestion.. We can certainly include this information in the paper for clarity.
>
> In terms of inference time, we collect the inference time of Rainbow and SD2-1 when generating 40 images per prompt, taking average of 60 prompts. We see that the average time per image for Rainbow is approximately 0.40 seconds, which is comparable to the standard diffusion model, SD2-1, which has an average inference time of 0.39 seconds. The forward time for the Rainbow module, that decompose original prompt representation into 40 prompt representations, itself is 0.104 seconds per prompt when generating 40 trajectories parallelly.
>
> Importantly, varying the number of trajectories (M) from 10 to 40 does not significantly affect the total generation time, as all the trajectories are predicted as output from the GFlownet at once. This parallel processing allows us to maintain efficient inference times, even when exploring a larger number of trajectories, without introducing substantial delays.
>
> We appreciate your suggestion and will ensure this information is clearly presented in the revised version of the paper. Thank you!
>
> -----------
> 3.  **When given a very specific prompt with little ambiguity (like a red circle on white background), does Rainbow still try to force diversity? How does it handle low-uncertainty conditions?**
>
> Thank you for your insightful comment.
>
>
> We have tried this prompt with both the SD2-1 baseline and Rainbow. Quantitatively, both models achieve comparable scores on Inception Score (IS), Visual Similarity (VS), and CLIP evaluation. Qualitatively, we observe that both models interpret the prompt creatively with red objects incorporating circular shapes, such as a red ball on a textured surface, a red circle appearing on a decorative plate, and red dots arranged in patterns against various backgrounds contains white color. Notably, none of the models produce images with a simple white background and a plain red circle.
>
> This is an interesting and helpful test case to understand models performance. We appreciate your insightful comment and will investigate similar prompts in the future.
>
> ----------
> 4.  **Are the interpretable edges (e.g. seasons) consistent across different training runs or did you select the best examples?**
>
>
>
> Yes, in different runs with varying model configurations and seeds, we can extract a consistent set of edges for features such as "seasons" and "devices" using the same edge frequency extraction method. The edge indices for each feature may differ across training seeds, but the presence of these interpretable features remains stable. This is because we do not predefine meanings for edges; instead, we allow the model to learn from the data during training. Consequently, the model assigns different sets of edges for features, but the underlying features themselves are reliably learned.
>
> Regarding to selecting samples, we generate and visualize 40 images for the natural images per model. We have provided all 40 generations in the appendix. For the visualization in Figure 2, we select images generated by Rainbow follow Spring-Summer-Autumn-Winter seasons.
>
> -----------------
> 5.  **Is there a principled way to choose M, N, and   ρ  for a new domain, or does it require extensive hyperparameter search -- not sure I feel the intuition here?**
>
> **Regarding hyperparameter selection for GFlowNet**, specifically for the parameters M, N, and ρ, our choices were based on several factors, including computational resources, the need for diversity, task plausibility, and results from fine-tuning.
> We adjusted the value of N by using a higher value for natural images to allow greater freedom in diversity and concept choices, while for medical datasets, we opted for a lower N value to balance diversity and ensure that the images still follow a domain-specific template. We maintained a relatively high graph sparsity, above 0.6; higher sparsity results in fewer sampled edges. This strategy enabled us to maximize the number of graphs given our available computational resources.
> We fine-tuned the combinations of M, N, and ρ for optimal performance in terms of both diversity and image quality. Notably, we observed that all combinations produced similar CLIP scores for prompt delivery evaluation. The fine-tuning results for chest X-rays are provided in Figure 11 in the appendix (referred to in line 313).
>
> We hope this answer clarifies your question.

---

> > ### Comment · Reviewer_75uy · 2025-08-03
> >
> > Thanks for your rebuttal, authors addressed my comments, and I prefer to save my rating.

---

> > > ### Author Response · Authors · 2025-08-06
> > >
> > > Thank you for your time and effort in reviewing my submission and rebuttal, as well as for maintaining your positive evaluation; we truly appreciate your support and feedback!

---

### Official Review · Reviewer_VBdx · 2025-07-04

**Clarity:** 3
**Significance:** 3
**Originality:** 3
**Rating:** 5
**Confidence:** 2

**Summary:**

This paper introduces a framework named Rainbow, designed to enhance the diversity of images produced by conditional generative models. The core problem it addresses is that a single input condition or prompt often has inherent uncertainty and can correspond to multiple plausible visual outputs, which standard generative models fail to capture, often producing repetitive results.

Rainbow is proposed as a model-agnostic layer that can be integrated with any pretrained conditional generative model, such as a Latent Diffusion Model (LDM), without altering its core components. The method operates in a multi-step process:

* Initial Representation: An input condition (e.g., a text prompt or a set of attributes) is first encoded into an initial latent representation using a standard condition encoder.

* Latent Graph Discovery with GFlowNets: The framework's key innovation is the use of Generative Flow Networks (GFlowNets). A GFlowNet is trained to sample a set of diverse trajectories over a latent graph structure. Each trajectory represents a different plausible interpretation of the uncertainty contained within the initial condition.

* Diverse Condition Generation: These diverse graph trajectories are then decoded into a set of distinct condition representations. Each new representation captures a unique aspect of the input condition's ambiguity.

* Diverse Image Generation: Finally, each of these diverse condition representations is fed into the frozen, pretrained generative model (e.g., an LDM) to produce a corresponding output image. The result is a set of varied images from a single initial prompt.

The authors evaluate Rainbow on both natural image (Flickr30k) and medical imaging (3D Brain MRI, Chest X-ray) datasets. The results are compared against baselines like Stable Diffusion and other diversity-enhancing methods using metrics for diversity (Vendi Score) and quality (Inception Score, FID). The experiments aim to show that Rainbow can generate a more diverse set of high-quality images that better reflect the inherent ambiguity of the input conditions.

**Questions:**

1. **Reward design** – The current GFlowNet reward is the exponential of the final denoising MSE, implicitly tied to modes the frozen LDM already renders well. Have you experimented with composite or alternative rewards (e.g., CLIP‐based semantic distances, pairwise contrastive terms) to reduce this potential bias, and if so how did they affect diversity and quality?

2. **Scaling the latent graph** – Reported models use up to *N = 20* nodes and *M = 40* trajectories. What practical or theoretical bottlenecks did you encounter when trying larger *N* or *M*, and do you foresee algorithmic modifications (e.g., sparse batching, trajectory pruning) that would let Rainbow scale to higher‐resolution video or multimodal inputs without prohibitive memory growth?

3. **Automated graph interpretability** – Qualitative edits (e.g., “season” edges, “support‐device” edges) suggest that some latent edges align with human concepts, yet Appendix C lists graph interpretation as future work. Could you outline planned or preliminary methods (attention tracing, concept activation maps, etc.) for systematically mapping edges to semantic attributes?

4. **Evaluation beyond automated metrics** – Diversity and fidelity are assessed with Vendi, IS, FID, and CLIP similarity. Are there plans to incorporate human studies or downstream‐task evaluations (e.g., radiologist scoring, prompt relevance surveys) to validate that Rainbow captures semantically meaningful uncertainty rather than pixel‐level variation?

5. **Clinical deployment pathway** – For the 3-D MRI and chest-X-ray settings, Appendix C notes the need for anatomical-plausibility checks before use in clinical pipelines. Could you elaborate on specific validation protocols (e.g., expert grading, quantitative anatomical metrics) and on whether constraints or priors could be integrated into Rainbow itself to enforce medical realism during sampling?

**Ethical Concerns:**

["NO or VERY MINOR ethics concerns only"]

**Final Justification:**

My initial score reflected concerns about the indirect reward signal and the lack of human-centric evaluation. The rebuttal addressed these issues by providing new, targeted experiments, including a test of an alternative reward function and a human study with expert evaluators. These additions have resolved my main reservations and strengthened the paper's claims. My score has been raised to recommend acceptance.

**Limitations:**

yes

**Quality:**

3

**Strengths And Weaknesses:**

## Strengths

*Methodological Novelty: The paper introduces a novel framework by coupling Generative Flow Networks (GFlowNets) with latent diffusion models. This approach, which samples multiple graph trajectories to capture multimodal uncertainty, is a notable contribution to the field of conditional image generation.

* Consistent Performance Across Domains: The method demonstrates strong and consistent empirical gains across three distinct domains: natural images, 3D brain MRIs, and 2D chest X-rays. The quantitative improvements in both diversity (Vendi Score) and fidelity (IS, FID) over strong baselines validate the effectiveness of the approach.

* Latent Graph Controllability: A significant strength is the demonstration that the learned latent graphs are not just a passive component but can be actively manipulated. The experiments showing that injecting "season-specific" or "support-device" edges leads to targeted semantic edits suggest that the graph encodes meaningful, editable concepts.

* Transparent Experimental Design: The paper provides a high degree of transparency, detailing hyperparameters and outlining the training algorithm with pseudocode. The inclusion of ablation studies on key parameters like N, M, and ρ further enhances the reproducibility and clarity of the work.

## Weaknesses

* Opaque Edge Semantics: While the latent graph is shown to be controllable, its components lack inherent interpretability. The mapping from graph edges to human-readable concepts is not automatic and must be inferred through post-hoc analysis. This "black box" nature of the graph's semantics is a core limitation, as it makes it difficult to understand or directly control the specific type of diversity being generated.


* Indirect Reward Signal for GFlowNet: The GFlowNet is trained to generate diverse graphs based on a reward signal derived from the final denoising error of the diffusion model. This is a highly indirect objective. The GFlowNet is not explicitly guided to find graphs that correspond to semantically meaningful variations, but rather graphs that result in low-error reconstructions. This could potentially bias the model toward modes of variation that are easy for the LDM to render, rather than ones that fully capture the true underlying uncertainty in the prompt.

---

> ### Author Rebuttal · Authors · 2025-07-31
>
> 1. **Reward design**: The current GFlowNet reward is the exponential of the final denoising MSE, implicitly tied to modes the frozen LDM already renders well. Have you experimented with composite or alternative rewards (e.g., CLIP‐based semantic distances, pairwise contrastive terms) to reduce this potential bias, and if so how did they affect diversity and quality?
>
>
>
>
> Thank you for your comment regarding reward design.
>
> We acknowledge that incorporating diversity and quality evaluation into the reward is a promissing approach. However, utilizing metrics such as VS, IS, or CLIP poses challenges, as these metrics require evaluating generated images using full-step diffusion, while our training typically employs only one diffusion step. Implementing training with full diffusion steps would significantly increase the time and memory requirements, making it challenging to effectively incorporate these rewards.
>
> We have conducted your suggestion on the pair-wise contrast (PWC) reward and provide a comparison between the original Rainbow model, which uses a reward based on the exponential of the Mean Squared Error (MSE) of the noise, and an updated version that incorporates pair-wise contrast into the reward function. We calculate the reward based on the exponential of two components: the negative MSE of the noises and the negative pair-wise MSE of the decoded trajectories.
>
> Quantitatively, compared to the original Rainbow model, the version with the additional PWC reward achieves comparable image quality as indicated by the IS score, higher diversity as reflected in the VS score, but shows a significant decrease in CLIP score, as represented in the table below. It's important to note that both VS and IS scores are computed without considering the text prompt.
>
>
> Intuitively, the introduction of the PWC reward aims to encourage greater diversity among the decoded trajectories. However, this emphasis on rewarding larger distances may lead to confusion in the model, as it has to balance the diversity of the generated outputs while maintaining contextual similarity.
>
> | | IS | VS | CLIP | Avg. inference time/image |
> |----------           |----------|----------|----------|----|
> | SD2-1                  | 8.49    | 25.63   | 30.27   |  0.39s  |
> | **Rainbow - original** | **10.45**| **28.90** | **30.32**  | 0.40s   |
> | Rainbow - PWC       | 9.43   | 29.81   | 27.47   |  0.40s  |
>
>
>
> Qualitatively, we observe two key trends: first, the pairwise MSE of decoded trajectories under the PWC reward increases significantly compared to the original version. Second, images generated using the PWC approach sometimes include irrelevant objects or omit concepts mentioned in the input prompt. For instance, when prompted with "Sunset scene with mountains," the generated images may lack the mountains and sky, featuring only greenery or even introducing unrelated elements like an office.
>
> While we agree that integrating PWC is a promising approach, we recognize that further investigation and fine-tuning are necessary to effectively utilize and control this method.
>
>
>
>
>
>
> Thank you for your insightful suggestion!
>
>
> -------
>
> 2. **Scaling the latent graph** – Reported models use up to  _N = 20_  nodes and  _M = 40_  trajectories. What practical or theoretical bottlenecks did you encounter when trying larger  _N_  or  _M_, and do you foresee algorithmic modifications (e.g., sparse batching, trajectory pruning) that would let Rainbow scale to higher‐resolution video or multimodal inputs without prohibitive memory growth?
>
>
>
> **Regarding hyperparameter selection for GFlowNet**, specifically for the parameters M, N, and ρ, our choices were based on several factors, including computational resources, the need for diversity, task plausibility, and results from fine-tuning.
> We adjusted the value of N by using a higher value for natural images to allow greater freedom in diversity and concept choices, while for medical datasets, we opted for a lower N value to balance diversity and ensure that the images still follow a domain-specific template. We maintained a relatively high graph sparsity, above 0.6; higher sparsity results in fewer sampled edges. This strategy enabled us to maximize the number of graphs given our available computational resources.
> We fine-tuned the combinations of M, N, and ρ for optimal performance in terms of both diversity and image quality. Notably, we observed that all combinations produced similar CLIP scores for prompt delivery evaluation. The fine-tuning results for chest X-rays are provided in Figure 11 in the appendix (referred to in line 313).
>
> **Regarding scaling to higher-resolution video or multimodal inputs**, **the inference time is nearly the same between Rainbow and the base model such as SD2-1**, as the burden is on the generation part, such as the UNet. The primary difference in inference time lies in the additional time required for Rainbow to decompose the input into multiple representations, rather than in the image generation phase itself. And even when we scale the number of images to be generated, the total inference time remains the same, as the forward time of the Rainbow module to generate representations in parallel is the same with M = 40, 20, or 10—around 0.104s/prompt.
>
> For instance, given the same computational resources, if we can generate 10 high-resolution images in parallel in a batch with a baseline model like Stable Diffusion, we can achieve the same with Rainbow using the same generative model. In scenarios where the available computational resources limit us to generating fewer images per batch at a time, e.g., a batch of 2, the SD baseline needs to generate images 5 times with a batch of 2 to get 10 images; Rainbow can still process multiple decomposed condition representations in parallel in a batch of 10, and then concurrently generate images in a batch of 2.
>
> Hope these answers clarify your questions.
>
> -----------
> 3. **Automated graph interpretability** Appendix C lists graph interpretation as future work. Could you outline planned or preliminary method?
>
> Thank you for your question regarding automated graph interpretability.
>
> First, we believe that computation on the latent space is crucial in aiding diversity and understanding the relationships between data points. By effectively interpreting the latent space, we can enhance the model's capacity to generate diverse outputs that still align with desired characteristics.
>
> Second, the following are some hybrid approaches to improve interpretability in this context:
>
>
>     1.  Pre-defined Clusters: We could establish pre-defined clusters and train the model on specific groups of data. This could involve focusing on specific categories, such as images containing people, images with animals in the prompts, or MRIs corresponding to specific age ranges. This targeted training could provide clearer insights into how different clusters influence the generated outputs.
>
>     2.  Integration with Domain-Specific Knowledge Graphs: We could train the latent graph alongside a domain-specific knowledge graph derived from the dataset. For instance, if a specific node in the predefined knowledge graph is activated, we can open relevant edges in the latent graph that connect to corresponding data points. This would create a more structured approach, allowing for the incorporation of prior knowledge into the model's decision-making process.
>
>
> By employing these strategies, we aim to enhance the interpretability of our latent graphs, providing more meaningful insights into how different features and relationships within the data contribute to the overall performance of the model.
>
> Hope this help with your consideration.
>
>
> ---------------
> 4. **Evaluation beyond automated metrics** Are there plans to incorporate human studies or downstream‐task evaluations (e.g., radiologist scoring, prompt relevance surveys) to validate that Rainbow captures semantically meaningful uncertainty rather than pixel‐level variation?
>
> **To address your concerns regarding human evaluation**, we present here the results of our human evaluation on the generated chest X-rays. In our study, we conducted human evaluations using 8 prompts, with 12 images generated per prompt, and invited three expert evaluators, all of whom are medical doctors, assessed the realism of the images by selecting which model—either the baseline SD or Rainbow—exhibited higher realism based on the given prompt descriptions. The results showed that **75% of the evaluators voted in favor of the Rainbow model**.
>
> We are expanding the human evaluation across all three datasets in our future work, including increasing the number of evaluators involved.
>
> --------------
> 5. **Clinical deployment pathway** Could you elaborate on specific validation protocols (e.g., expert grading, quantitative anatomical metrics) and on whether constraints or priors could be integrated into Rainbow itself to enforce medical realism during sampling?
>
> Regarding validation after training, we can conduct human evaluation to select the best model, as mentioned in the answer to question 4.
>
> Regarding priors to integrate into Rainbow training and benefit the sampling results, one approach could be to utilize predefined domain-specific causal graphs for the effects of age and sex on brain MRI, or diseases on chest X-Ray. Another approach could integrate ideas from our response to question 3.
>
> Thank you for your insightful comment.

---

> > ### Comment · Reviewer_VBdx · 2025-08-06
> >
> > Thank you for the rebuttal and the new experimental results.
> >
> > The new experiment on reward design and the human evaluation study for the chest X-ray generations have addressed several of the initial questions. The clarifications regarding scalability and interpretability were also noted.
> >
> > The rebuttal has sufficiently addressed the initial concerns to warrant an increase in score

---

> > > ### Author Response · Authors · 2025-08-06
> > >
> > > Thank you for your time and effort in reviewing my submission and rebuttal, as well as for increasing the score; we truly appreciate your recognition and support!

---

### Note · Authors · 2025-08-12

Thank you to all reviewers and AC members for your efforts reviewing our submission with insightful suggestions.

### **Summary of Rainbow**:

Rainbow introduces a novel approach for diverse conditional image generation, addressing cases where the input condition contains uncertainty, leading to multiple plausible outputs. Rainbow is to decompose the uncertain input condition into diverse representations, each capturing a specific aspect of the uncertainty.

**Key innovations include:**

-   First, we integrate a graph structure into condition representation learning and capture uncertainty of the given condition by sampling multiple trajectories over the graph.
-   Next, we use GFlowNet algorithm, which sample multiple possible graphs for tasks with uncertainty, to sample diverse trajectories. Thus, the uncertainty of the given input is captured.
-   Finally, each trajectory is decoded into a condition representation and passed through a frozen diffusion model to generate one output image.
-   Experiments on natural images, 3D brain MRIs, and Chest X-rays proved Rainbow's ability to generate diverse plausible outputs by outperforming baselines in both diversity (VS score) and quality (IS, FID) evaluation while adhering input prompt (CLIP).

---

### **Main points of Author-Reviewers Discussion**:

1.  **Inference time and scalability of Rainbow**. Rainbow's generation time is relatively the same as baseline and easy to scale, as the parallel condition decomposition time is small and the same when decomposing into 10 or 40 representations.

2.  **Hyperparameters** We have provided detailed finetuning processes, addressing this concern.

3.  **Edge interpretability and consistency**. We can extract edge meanings ("season" edges) by automatically analyzing edge frequency. This extractable concepts are consistent across runs.

4.  **Human evaluation** In the rebuttal, we provided medical experts evaluation that preferred Rainbow for 75% realism over the baselines.

5.  **Prompt adherence** In the submission, Rainbow achieves similar CLIP scores compared to the baselines, demonstrating that Rainbow still adheres the input prompt.

6.  **Baselines** In the rebuttal, we added a comparison to DDIM for natural images. Rainbow outperformed the DDIM in both quantitative and qualitative results.

---

Overall, reviewers were all satisfied with the rebuttal.

We hope this summary will help the ACs and reviewers with the paper decision!

---

### Decision · Program_Chairs · 2025-09-17

**Decision:**

Accept (poster)

**Comment:**

The paper proposes a method for generating diverse images from a fixed trained conditional generative model, by decomposing the conditioning signal into diverse latents, and then integrating graphs over these latents into prompt representations using GFlowNets.

The method achieves good results on both natural and medical images, is new, and creative.

There were some concerns around baselines, performance, inference time, and presentation, but these were addressed satisfactorily in the author/reviewer discussion.

Overall, all the reviewers are positive about the paper, I agree and recommend acceptance.